# The receptor kinase SRF3 coordinates iron-level and flagellin dependent defense and growth responses in plants

Matthieu Pierre Platre [1], Santosh B. Satbhai [1,2,3], Lukas Brent[1], Matias F. Gleason[1], Min Cao[1], Magali Grison [4], Marie Glavier [4], Ling Zhang[1], Christophe Gaillochet[1], Christian Goeschl[2], Marco Giovannetti [2], Balaji Enugutti [2], Julie Neveu[5], Marcel von Reth[6], Ruben Alcázar[6], Jane E. Parker [6,7], Grégory Vert [5], Emmanuelle Bayer [4] & Wolfgang Busch [1,2] ✉

Iron is critical for host–pathogen interactions. While pathogens seek to scavenge iron to spread, the host aims at decreasing iron availability to reduce pathogen virulence. Thus, iron sensing and homeostasis are of particular importance to prevent host infection and part of nutritional immunity. While the link between iron homeostasis and immunity pathways is well established in plants, how iron levels are sensed and integrated with immune response pathways remains unknown. Here we report a receptor kinase SRF3, with a role in coordinating root growth, iron homeostasis and immunity pathways via regulation of callose synthases. These processes are modulated by iron levels and rely on SRF3 extracellular and kinase domains which tune its accumulation and partitioning at the cell surface. Mimicking bacterial elicitation with the flagellin peptide flg22 phenocopies SRF3 regulation upon low iron levels and subsequent SRF3-dependent responses. We propose that SRF3 is part of nutritional immunity responses involved in sensing external iron levels.

Iron is a critical micronutrient for all living organisms. While iron is very abundant in the Earth's crust, its bioavailability is low. Organisms have evolved efficient iron uptake mechanisms that include a variety of membrane-associated systems that absorb iron unbound or bound to iron-binding molecules[1]. Mammals acquire iron mainly through the glycoprotein transferrin while bacteria, fungi and plants have evolved diverse systems that include siderophores, which are small, high-affinity diffusible secondary metabolites that chelate $Fe^{3+}$ from the surrounding environment[1]. In plants, Graminaceae species employ plant specific siderophores while non-Graminaceae such as *Arabidopsis thaliana* depend on an iron reduction-based uptake strategy[2].

During pathogen attack, iron is at the nexus of host–pathogen interaction as both organisms compete for this metal. Pathogens scavenge iron from the host through siderophore secretion while the host aims to sequester iron to prevent pathogen virulence. Thus, host external iron sensing and internal iron homeostasis regulation are of particular importance to prevent pathogen infection, and are part of the first line of defense called nutritional immunity[3].

[1]Salk Institute For Biological Studies, Plant Molecular and Cellular Biology Laboratory, 10010 North Torrey Pines Road, La Jolla, CA 92037, USA. [2]Gregor Mendel Institute (GMI), Austrian Academy of Sciences, Vienna Biocenter (VBC), Dr. Bohr-Gasse 3, 1030 Vienna, Austria. [3]Department of Biological Sciences, Indian Institute of Science Education and Research, MohaliSector 8, SAS NagarPunjab 14046, India. [4]University of Bordeaux, CNRS, Laboratoire de Biogenèse Membranaire, UMR 5200 CNRS, 33140 Villenave d'Ornon, France. [5]Plant Science Research Laboratory (LRSV), UMR5546 CNRS/Université Toulouse 3, 24 Chemin De Borde Rouge, 31320 Auzeville-Tolosane, France. [6]Department of Plant-Microbe Interactions, Max Planck Institute for Plant Breeding Research, Cologne, Germany. [7]Cologne-Düsseldorf Cluster of Excellence on Plant Sciences (CEPLAS), 50225 Düsseldorf, Germany. ✉e-mail: wbusch@salk.edu

In mammals, two receptors, transferrin receptor 1 and 2 (TfR), which bind extracellular transferrin-associated iron, play a major role in regulating external iron sensing and homeostasis[3]. Upon host–pathogen interaction, bacterial siderophores outcompete the host iron-bound to transferrin, which in turn leads to a loss of iron triggering independent local and systemic responses in the host[4]. Locally, the loss of iron induces TfR endocytosis and intracellular iron storage via ferritins. Systemically, TfR activation triggers stimulation of the BMPR complex to increase the expression of iron uptake genes[4]. The latter response is intertwined with the defense pathways since the inflammatory Interleukine-6 pathway directly interacts with the BMPR complex to regulate iron uptake genes[4]. In *Drosophila melanogaster*, Transferrin-1 was recently shown to activate NF-κB, toll and immune deficiency immunity pathways, thereby mediating nutritional immunity through the control of intracellular iron partitioning[5].

Although flowering plants do not contain TfR in their genomes[6], iron homeostasis and defense responses are linked[7]. Here, FERRETINS (FER) and NATURAL RESISTANCE-ASSOCIATED MACROPHAGE PROTEIN (NRAMPs) were shown to be involved in iron sequestration upon pathogen attack[8,9]. Moreover, the metal transceptor IRON-REGULATED TRANSPORT 1 (IRT1) is critical to mount efficient defense responses[10]. Transcriptional signatures of *Pseudomonas simiae* WCS417 and long-term iron deficiency in leaves display an overlap of about 20%, among these genes, and the transcription factor MYB DOMAIN PROTEIN 72 (MYB72) plays a role at the interface of both signaling pathways[11]. Recently, a protein effector from the foliar pathogen *Pseudomonas syringae* was shown to disable a key iron homeostasis regulator, the E3 ligase BRUTUS (BTS), to increase apoplastic iron content and promote colonization[12]. Finally, the presence of the microbial siderophore, deferrioxamine (DFO), affects the transcriptional landscape of iron homeostasis and immunity genes, suggesting a role for siderophores in mediating nutritional immunity[10].

While the link between iron deficiency and immunity has been well documented in plants, the mechanism by which iron concentrations are sensed, and how they impinge on iron homeostasis, defense and growth pathways are unknown. Here, we identify the leucine-rich repeats receptor kinase STRUBBELIG RECEPTOR KINASE 3 (SRF3) as a regulator of root growth under low iron levels in *Arabidopsis thaliana*. We find that root growth is rapidly reduced upon encountering low iron levels and modulates root iron homeostasis in a *SRF3*-dependent manner. The regulatory capacity of SRF3 is dependent on its kinase and extracellular domains. Both domains are required for SRF3 partitioning between the plasmodesmata and the so-called "bulk PM" where it acts as a negative regulator of callose synthases and is degraded upon low iron conditions in both sub-populations. We further establish that SRF3 is a molecular link between low external iron levels and bacterial defense responses, as SRF3 is required to mediate root immune pathways to the flagellin peptide flg22 by the same mechanisms used for its response to low iron conditions. Our work uncovers a close coordination of responses to low iron levels and immunity pathways which indicates that *SRF3* is located at the nexus of both pathways, thereby constituting a key player in plant nutritional immunity.

## Results

### SRF3 is a regulator of iron homeostasis genes and root growth under low iron levels

Genome wide association studies (GWAS) for root growth rate under low iron levels revealed multiple significantly associated single-nucleotide polymorphisms (SNPs; Supplementary Fig. 1a, b and Supplementary Data 1). The association peak containing most significant SNPs was observed on chromosome 4 near the genes AT4G03390 (*STRUBBELIG-RECEPTOR FAMILY 3, SRF3*) and AT4G03400 (*DWARF IN LIGHT 2, DFL2*) for root growth between days 4–5 (Fig. 1a and Supplementary Data 1). To identify potential causal genes at this locus, we obtained Col-0 T-DNA mutant lines for these genes (Supplementary Fig. 1c, d). We then measured the root growth of these lines grown on iron sufficient and low iron medium. To provide more clarity, we calculated for each genotype the root growth response (the ratio of three days of root growth in low iron and sufficient iron conditions). While the *dfl2* T-DNA mutant lines responded similarly to wildtype (WT) to low iron levels, *srf3* T-DNA lines displayed a significantly decreased root growth response compared to WT when exposed to different iron levels using the iron chelator ferrozine (−FeFZ) (Fig. 1b, c and Supplementary Fig. 1e–i). This indicated that *srf3* roots are insensitive to these conditions. Moreover, *srf3* mutants showed a slight reduction in their root growth rate in iron sufficient conditions compared to WT and responded similarly to WT to iron excess conditions (Supplementary Fig. 1g–i). Overall, our data show that *SRF3* is required for an appropriate root growth response to low iron levels.

To explore the impact of *SRF3* on iron homeostasis, we performed RNAseq on roots from two independent *srf3* alleles and WT under iron sufficient growth conditions. Several key iron homeostasis regulators (*BTS, BHLHO39, PYE*) and iron compartmentalization-related genes that are involved in iron transport to the vacuole (*ZIF1*) were upregulated in *srf3* mutants while a key iron distribution transporter involved in shoot-to-root iron partitioning was downregulated (*NAS4*; Fig. 1d and Supplementary Data 2). Consistent with a mis-regulation of iron responsive genes, the transcriptional reporter line of the low iron inducible iron transporter *IRT1 (pIRT1::NLS-2xYPet)* in *srf3-4* mutant showed a decreased activation after 24 h under low iron (Fig. 1e). Staining for iron with dyes revealed that *srf3* mutants accumulate more iron compared to the WT, thereby phenocopying *bts-1* and *opt3-2*, two mutants known to accumulate ectopic iron[13,14] (Fig. 1f and Supplementary Fig. 2a–d). Importantly, the increased iron levels in *srf3-3* do not stem from a misdistribution of iron in the seeds since iron localization was not altered in *srf3* mutant seed embryos compared to WT but different from the positive control *vit-1*[15] (Supplementary Fig. 2e). Taken together, these results indicate that *SRF3* is a post-embryonic regulator of iron homeostasis genes.

Since we had shown the function of SRF3 in regulating root growth responses to low iron levels and iron homeostasis, we further investigated the allelic variation at the *SRF3* locus and analyzed accessions according to the pattern of the four top marker polymorphisms associated with the growth response under low iron conditions. Accordingly, the haplogroup A grows slowly on low iron medium and the haplogroups B, C, D grow faster (Supplementary Fig. 3a–c). While the haplogroup A and haplogroup B differed from several candidate polymorphisms including a larger deletion in the promoter region (Supplementary Fig. 3a), they do not show any significant differences in *SRF3* transcript level accumulation under sufficient and low iron levels (Supplementary Fig. 3d). To test *SRF3* allelic variation in controlling root growth under low iron, we expressed the *SRF3* full genomic sequence starting 1688 pb upstream of 5′UTR, which unfortunately does not include the deletion, from two slow and three fast growers in *srf3-4* mutant and quantified the root growth response. We did not observe a clear correlation between the sequences of the fast or slow growers and the root growth rate under low iron levels (Supplementary Fig. 3e). Taken together, our data show that *SRF3* is a negative root growth regulator under low iron levels and is involved in the post-embryonic regulation of iron homeostasis. While tempting to speculate that SRF3 variants and in particular the deletion might be involved in determining root growth rate in low iron conditions in accessions, its roles as well as those of other close by genes need further investigation. Because, to our knowledge no other LRR-RK had been directly implicated in the response to low iron levels, we focused on characterizing the signaling and molecular roles of SRF3 in this response.

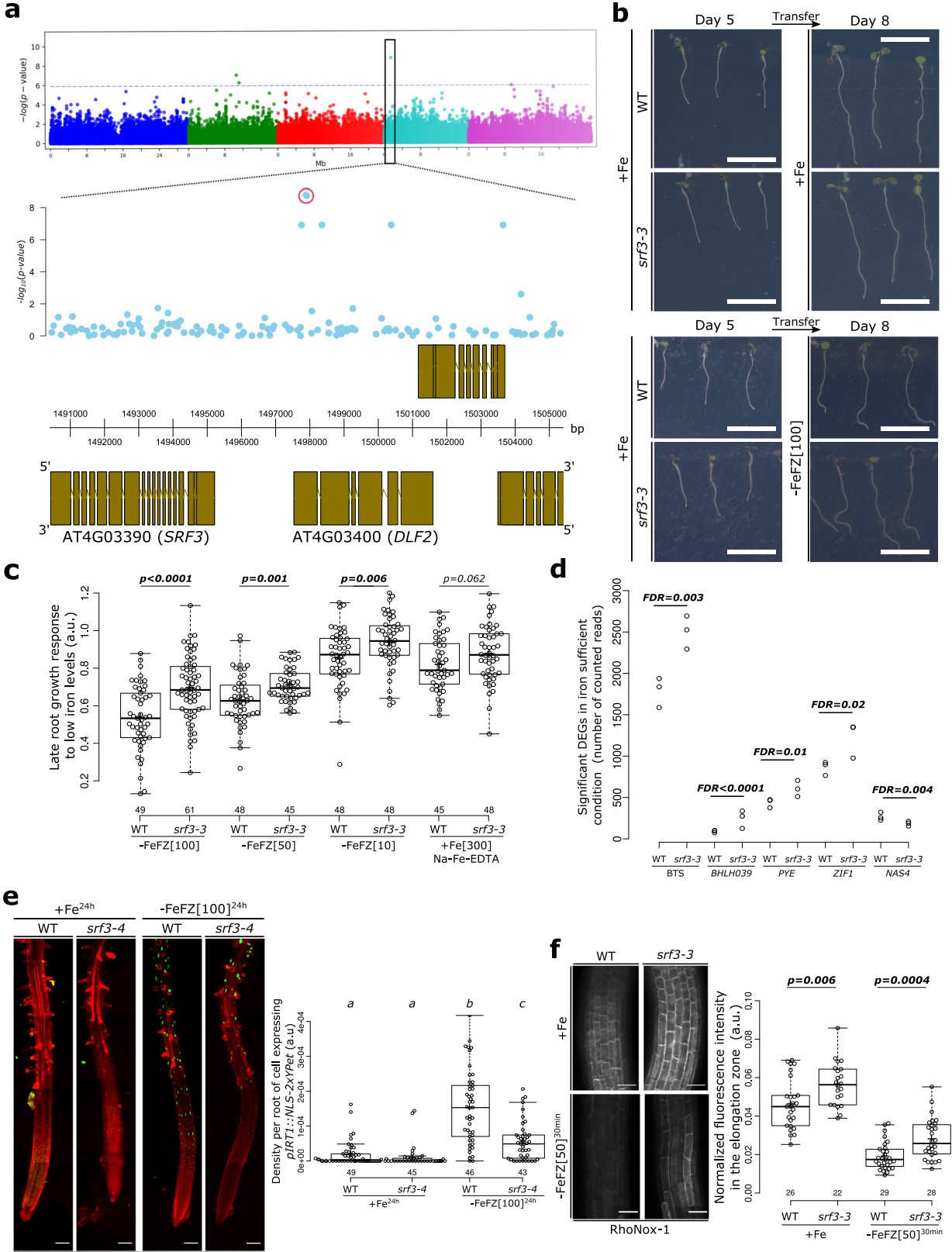

## The early growth response to low iron is dependent on SRF3 protein levels at the plasma membrane

We next wanted to elucidate the mechanism of SRF3 in regulating growth in response to low iron conditions. First, we set out to characterize the *SRF3* expression pattern using transcriptional and translational reporter lines in roots. A SRF3-2xmCHERRY fusion construct driven by its own promoter (2432 pb upstream of upstream of 5'UTR including the region that was deleted in the fast-growing accessions) fully complemented the *srf3-3* root growth defect under low iron levels, showing the functionality of the construct (Supplementary Figs. 1f and 4a). Analysis of the transcriptional reporter line revealed that the *SRF3* promoter is active in the differentiation and elongation

**Fig. 1 | SRF3 regulates root growth and iron homeostasis upon low iron conditions. a** Upper panel: Manhattan plot for GWA mapping of the root growth rate day 4–5 of natural accessions grown under low iron conditions. The horizontal dash dot line corresponds to a 5% false discovery rate threshold. Black box indicates the significantly associated SNP that is in proximity to *SRF3 (At4g03390)*. Lower panel: Magnified associations in the *SRF3* region with gene models. **b** Representative images of 5 day old seedlings of WT and *srf3-3* under iron sufficient medium for 5 days (left panel) and then transferred to iron sufficient media (+Fe; upper right panel), or to low iron medium (−FeFZ 100 μM, lower right panel) and grown for 3 days. This phenotype has been observed at least 12 times. Scale bars, 1 cm. **c** Boxplots of late root growth response of plants grown for 3 days on different iron levels (−FeFZ 10, 50, 100 μM or Na-Fe-EDTA 300 μM) in WT and *srf3-3* seedlings [two-ways student test ($p < 0.05$), n.s. non-significant]. **d** RNAseq read counts of differentially expressed iron homeostasis genes in roots of WT and *srf3-3* in iron sufficient conditions. **e** 5 day old seedlings stained with propidium iodide (PI; red channel) expressing *pIRT1::NLS-2xYPet* (green channel) in WT and *srf3-4* on sufficient (+Fe) or low (−Fe) iron medium and the related quantification [one-way ANOVA follows by a post hoc Tukey HSD test, letters indicate statistical differences ($p < 0.05$)]. Scale bars, 100 μm. **f** Confocal images of 5 day old seedlings stained with RhoNox-1 in WT and *srf3-3* on sufficient medium (+Fe; upper panel) or low iron medium (ferrozine 50 μM, 30 min; lower panel) and related quantification [Independent two-ways student test ($p < 0.05$)]. Scale bars 50 μm.

zones, and to a lesser extent in the transition zone (Fig. 2a). The SRF3 fluorescent protein fusion reporter line was detected mainly at the PM in the apical and basal meristem and to a lesser extent in the transition, elongation, and differentiation zones in Col-0 and Landsberg *erecta* (Ler-0) WT backgrounds (Fig. 2a and Supplementary Fig. 4b–f). Moreover, line expressing SRF3-mCITRINE under the control of the ubiquitous promotor *UBIQUITIN10* (*UBQ10*) showed the same expression pattern as observed in lines with SRF3 driven by its native promotor (Supplementary Fig. 4g). Because of the divergence of SRF3 transcript and protein abundance, we reasoned that the SRF3 protein and/or transcript might be cell-to-cell mobile, or that *SRF3* might be expressed only transiently in the meristematic cells. Analysis of numerous root tips showed that some roots expressed the *SRF3* transcriptional reporter in the meristematic zone, a finding also supported by the analysis of single-cell RNAseq data[16] (Supplementary Fig. 4h–k). However, this does not exclude the possibility of cell-to-cell transport. Overall, our data show that *SRF3* is constantly transcribed and translated in the transition-elongation zone and transiently or only in a subset of cells in the meristematic zone.

*SRF3* encodes a gene belonging to the protein family of leucine-rich repeats receptor kinases (LRR-RKs) that are known to be involved in early signal transduction[17]. We hypothesized that *SRF3* might mediate a novel, immediate root response to changes in external iron levels. Observing root growth via live-light transmission microscopy for 12 h, we found that low iron levels elicited a significant decrease of root growth after 4 h and that this response was abolished in *srf3* mutants (Fig. 2b and Supplementary Fig. 4l and Movies 1–2). We next tested whether the observed root growth response to low iron was primarily due to changes in the meristem size and/or cell elongation. While meristem length did not significantly change in WT upon low iron, elongated cells were significantly shorter (Supplementary Fig. 4m, r). Consistent with the expression pattern of *SRF3*, the unresponsiveness in *srf3* mutants to low iron is explained by a lack of restriction of elongation (Supplementary Fig. 4q, r). We next tested whether SRF3 protein abundance or *SRF3* transcription are altered in response to low iron conditions in the elongation zone. The signal intensity in the *SRF3* transcriptional reporter line did not differ between the two iron regimes and was similar in the control line H2B-mSCARLET (Supplementary Fig. 5a). However, the fluorescent signal intensity in a reporter line in which a SRF3 reporter fusion protein was driven by *UBQ10* promoter (SRF3$^{WT}$) or its native promotor significantly decreased at the PM under low iron treatment compared sufficient and iron and to Lti6b control line (Fig. 2c and Supplementary Fig. 5b). In a time-lapse experiment, a signal decrease was recorded after 50 min in SRF3$^{WT}$ but not in the other lines under low iron levels without the addition of ferrozine (referred as −Fe) compared to iron sufficient condition (Fig. 2d and Supplementary Movies 3, 4, and 5). Moreover, a western blot for SRF3 fused to mCITRINE using anti-GFP showed that low iron levels without ferrozine for 4 h also led to a decrease of SRF3 protein in the root (Supplementary Fig. 5c). We next set out to dissect the role of the functional domains of SRF3 by generating a truncated version of SRF3 in which the

extracellular domain had been removed (SRF3$^{ΔExtraC}$) and a kinase dead version (SRF3$^{KD}$, Supplementary Fig. 5d, e). While the SRF3 protein levels were decreased in the line with the functional protein (SRF3$^{WT}$) after 2 h of exposure to low iron conditions, they were not for SRF3$^{ΔExtraC}$ or SRF3$^{KD}$ lines (Fig. 2c). This shows that both, the extracellular cellular and kinase domains are required to mediate the decrease of SRF3 protein at the PM in response to low iron levels.

We then investigated whether SRF3 levels control early root growth rate under low iron conditions. Much like *SRF3* loss of function, constitutive expression of *SRF3* abolished the early root growth response to low iron levels (Fig. 2e). However, we observed an opposite effect in *srf3* mutant and SRF3$^{WT}$ overexpressing plants during the late response to low iron (Supplementary Fig. 5f, g). This highlights that early and late root growth responses to low iron levels are distinct from each other. Even though we cannot fully explain these opposite phenotypes in SRF3 overexpressing lines, which most likely due to stoichiometry change impairing its proper functionality, we used this property to interrogate SRF3 domain functions. To do so, we investigated the early growth response of the overexpressing lines of SRF3$^{ΔExtraC}$ (*pUBQ10::SRF3$^{ΔExtraC}$-mCITRINE/Col-0*), SRF3$^{KD}$ (*pUBQ10::SRF3$^{KD}$-mCITRINE/Col-0*) to low iron conditions compared to SRF3$^{WT}$ (*pUBQ10::SRF3-mCITRINE/Col-0*). In contrast to the SRF3$^{WT}$ version, we observed that roots of SRF3$^{ΔExtraC}$ and SRF3$^{KD}$ presented a phenotype close to WT plants (Fig. 2e and Supplementary Fig. 5f, g). As all the overexpression fusion proteins were expressed in the WT background, this indicates that the mutated SRF3 versions are not functional to mediate root responses to low iron levels. Altogether, our results suggest that the root growth response to low iron conditions requires degradation of SRF3 at the PM, which is dependent on the extracellular and kinase domains. However, we cannot directly conclude that the phenotype was due to a lack of SRF3 kinase activity since the point mutation in the kinase domain also impaired its degradation, which might be required for root growth regulation.

**SRF3 resides in the plasmodesmata and bulk plasma membrane regions where its level is decreased under low iron conditions**
During the analysis of SRF3 expression, we had noticed its enrichment at the PM with an apical-basal localization in punctate foci but also along the entire PM, referred to as bulk PM (Supplementary Fig. 6a–c). We tested the role of its extracellular domain and kinase domain for specifying its heterogenous distribution by calculating the standard deviation of the mean intensity (SDMI) along the apical-basal side of PM. In standard conditions, we found that compared to the WT version, SRF3$^{ΔExtraC}$ and SRF3$^{KD}$ showed a strong and slight decrease of the SDMI, respectively, while the removal of the kinase domain (SRF3$^{ΔKinase}$) did not lead to any changes (Fig. 3a and Supplementary Fig. 6b). A similar trend was observed when looking at the apical-basal polarity (Fig. 3a). This indicates that the extracellular domain is necessary and sufficient to drive SRF3 into the PM-associated foci and might suggest a direct role of the kinase domain in its partitioning. Upon low iron levels, no changes were observed for SRF3$^{ΔExtraC}$ and SRF3$^{KD}$ as well as in the control line, Lti6b while a decrease of SDMI and polarity were

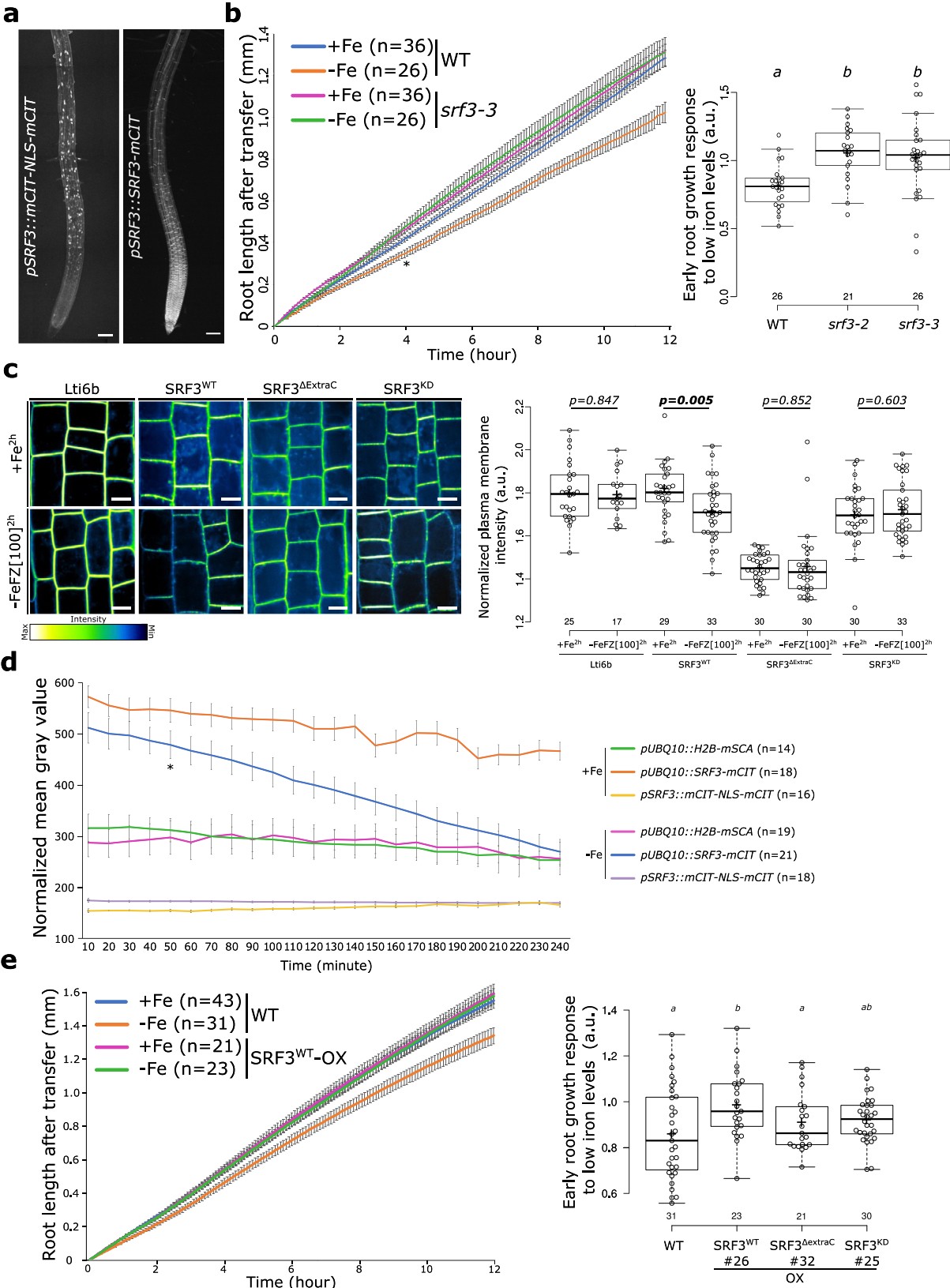

observed for SRF3[WT] (Fig. 3a and Supplementary Fig. 6c). This points to a major role of the extracellular domain and putatively to a lesser extent to the kinase domain in the removal of SRF3 from the foci upon exposure to low iron levels.

We then addressed the nature of the PM-associated punctuated structures. Analysis of the intensity distribution profile of SRF3 with

PM structure marker lines revealed a specific co-localization of SRF3 with plasmodesmata-associated proteins CALS3 and PDLP3 but not the general PM marker Lti6b (Fig. 3b). In accordance with the SDMI results, SRF3[WT] but not SRF3[ΔExtraC] co-localized with signals from aniline blue staining that stains β-1,3-glucans that are particularly enriched in plasmodesmata (PD, Supplementary Fig. 6d). This strongly indicated

**Fig. 2 | SRF3 regulates early root growth response and undergoes degradation upon low iron levels. a** Maximum intensity projection of confocal images of root tips of 5 day old seedlings expressing *pSRF3::mCITRINE-NLS-mCITRINE* and *pSRF3::SRF3-mCITRINE in Col-0*. This pattern has been observed at least hundreds of times. Scale bars, 100 µm. **b** Graph showing time-lapse analysis for 12 h of the root length of WT and *srf3-3* under sufficient (+Fe) and low (−Fe) iron media [error bars indicate standard error of the mean (SEM); Asterix: first timepoint with significant difference between WT in +Fe and −Fe conditions according to a mixed effect model (*p* < 0.05)] and the related quantification including the *srf3-2* mutant [ANOVA with post hoc Tukey test; Letters: statistical differences (*p* < 0.05)]. **c** Confocal images of root epidermal cells in the elongation zone of 5 day old seedlings expressing *p35s::eGFP-Lti6b/col-0*, *pUBQ10::SRF3^WT^-mCITRINE/col-0*,

*pUBQ10::SRF3^ΔExtraC^-mCITRINE/col-0*, *pUBQ10::SRF3^KD^-mCITRINE/col-0* under iron sufficient (+Fe, 2 h) and low iron levels (−FeFZ 100 µM, 2 h) and the related quantification [two-ways student test (*p* < 0.05), n.s.: non-significant]. Scale bars 10 µm. Note that the pictures have been pseudo-colored in green fire blue to emphasize changes in fluorescence intensity. **d** Graph representing the fluorescence intensity in the root tip of the indicated protein fusions under sufficient (+Fe) and low (−Fe) iron media [error bars indicate SEM; Asterix: first timepoint with significant difference between +Fe and −Fe for *pUBQ10::SRF3-mCITRINE* according to a mixed effect model (*p* < 0.05)]. **e** Graph showing time-lapse analysis for 12 h of the root length of WT and *SRF3^WT^-OX* under sufficient (+Fe) and low (−Fe) iron media [error bars indicate SEM] and related quantification including SRF3^ΔExtraC^ and SRF3^KD^ [ANOVA with post hoc Fisher test; Letters: statistical differences (*p* < 0.05)].

---

that SRF3 is in close vicinity to the PD and that this is mediated through its extracellular domain. To confirm SRF3 subcellular dynamics at higher resolution, we conducted immunogold-labeling electron microscopy of the *pSRF3::SRF3-GFP* line using an anti-GFP antibody. In standard conditions, SRF3 signal was localized at the bulk PM and to the neck region of the plasmodesmata (Fig. 3c). Upon low iron conditions SRF3 was removed from the PD and the bulk PM, with the most drastic removal being observed at the PD (Fig. 3c). As a recent report had shown that some plasmodesmata-associated receptor kinases have a fast and reversible association between bulk PM and plasmodesmata under abiotic stress, which alters their diffusion rates within the PM[18], we estimated SRF3 diffusion via fluorescence recovery after photobleaching (FRAP). We found that a decrease of iron levels did not change SRF3 diffusion (Supplementary Fig. 6e), indicating that the decrease of SRF3 might not accompanied by a change in its partitioning. Taken together, our data indicate that SRF3 is associated with the bulk PM but also highly enriched at the neck of the plasmodesmata, in an extracellular domain-dependent manner. Under low iron, SRF3 becomes depleted mainly from the PD and to a lesser extent from the bulk PM, a process, which is dependent on both SRF3 functional domains.

## Early exposure to low iron mediates SRF3-dependent callose deposition without modifying cell-to-cell movement

Immunogold-labeling electron microscopy suggested that SRF3 is particularly concentrated at the PD neck, which is highly enriched in sterols[19]. Depleting plants expressing SRF3^WT^ of sterols using sterol inhibitors, Fenpropimorph (Fen) and Lovastin (Lova), confirmed the localization of SRF3 in the PD neck since a decrease of SRF3 polarity and SDMI were observed (Supplementary Fig. 7a). This region is critical for regulating cell-to-cell trafficking, as it is where callose turnover is thought to be regulated to tune plasmodesmata permeability[20]. Iron homeostasis depends on long- and local-distance signaling relying on cell-to-cell movement of signaling molecules to activate *IRT1*[21–23]. We therefore hypothesized that SRF3 might regulate cell-to-cell communication through callose turnover to properly activate *IRT1*. Immunofluorescence staining with a callose antibody indicated that low iron levels trigger callose deposition in the epidermis and cortex cells of WT root tips highlighting the influence of iron levels on callose deposition (Fig. 4a). In *srf3* mutants, we observed an increase of callose even in the basal condition while callose levels were not responsive to iron depleted media compared to WT (Fig. 4a). Our data therefore support that those early responses to low iron include an increased callose deposition and that SRF3 negatively regulates this process. To corroborate this finding, we used aniline blue to study callose deposition in the epidermis of the root elongation zone. In agreement with the antibody-based findings, low iron rapidly enhanced callose deposition in WT and was higher in the positive control CALS3-OX[24] (Supplementary Fig. 7b). However, increased callose was not observed in WT when adding 2-deoxy-d-glucose (DDG), a well-characterized callose synthase inhibitor (Supplementary Fig. 7b)[24,25]. In *srf3* mutants, no

difference in aniline blue signal intensity was observed under iron sufficient conditions while a significantly higher increase was observed under low iron compared to WT in the same condition (Fig. 4b). Although callose immunofluorescence staining and aniline blue slightly differed, both experiments suggest that callose is synthesized by callose synthases in 4 h after exposure to low iron conditions in an SRF3-dependent manner.

We next investigated the callose deposition pattern upon low iron levels using immunogold-labeling electron microscopy and anti-callose antibodies. In the *pSRF3::SRF3-GFP* line, where under low iron conditions we had observed the dual loss of SRF3 localization at the PD and bulk PM (Fig. 3c), we found a slight albeit not significant increase of gold particles at the PD under iron deprivation and no change at the PM (Fig. 4c). In light of the confocal microscopy-based analysis of callose antibody and AB staining results, which showed an increase of callose upon low iron conditions, we conclude that either the immunogold technique is not very sensitive to small changes or that callose was deposited at an unobserved part of the samples. We then tested whether cell-to-cell protein movement was perturbed in a SRF3-dependent manner. For this, we monitored the ability of GFP expressed in companion cells using *pSUC2::GFP* to diffuse to the surrounding cells through the plasmodesmata but observed no difference (Supplementary Fig. 7c)[24,26]. To corroborate this observation, we photoactivated DRONPA-s fluorescent protein in a single root epidermal cell and monitored its spread to the upper and lower surrounding cells[27]. We noticed a decrease of signal in the activated cell and a concomitant increase in the surrounding cells, resulting from cell-to-cell movement (Fig. 4d). Consistently, with *pSUC2::GFP* observations, no difference between conditions and/or genotypes was observed (Fig. 4d). Altogether, our results suggest that an early decrease of iron levels swiftly leads to SRF3-dependent modulation of callose deposition, which might involve callose synthases, however, this does not generally impede cell-to-cell movement.

## Iron homeostasis and root growth are steered by SRF3-dependent callose synthases

While *IRT1* activation is dependent on *SRF3* (Fig. 1e), this appears not to rely on a restriction of cell-to-cell movement via callose synthases-mediated callose deposition during the early responses to low iron conditions. We therefore reasoned that *IRT1* regulation might rely on early signaling events that are dependent on callose synthases, or that *IRT1* regulation only occurs at a later stage of the response. We first tested whether *IRT1* is regulated at the time during which SRF3-dependent callose deposition occurs. A 16-h time-lapse analysis of *IRT1* promoter activity indicated that it becomes active during the first hours of low iron conditions while no or little activity was observed in iron sufficient media (Fig. 5a and Supplementary Movies 6 and 7). In *srf3-4* mutant roots, we observed a lower expression of the *IRT1* reporter line upon low iron conditions compared to WT, indicating that early *IRT1* transcriptional activation depends on *SRF3* (Fig. 5b). Next, we tested whether callose synthases activity was important to

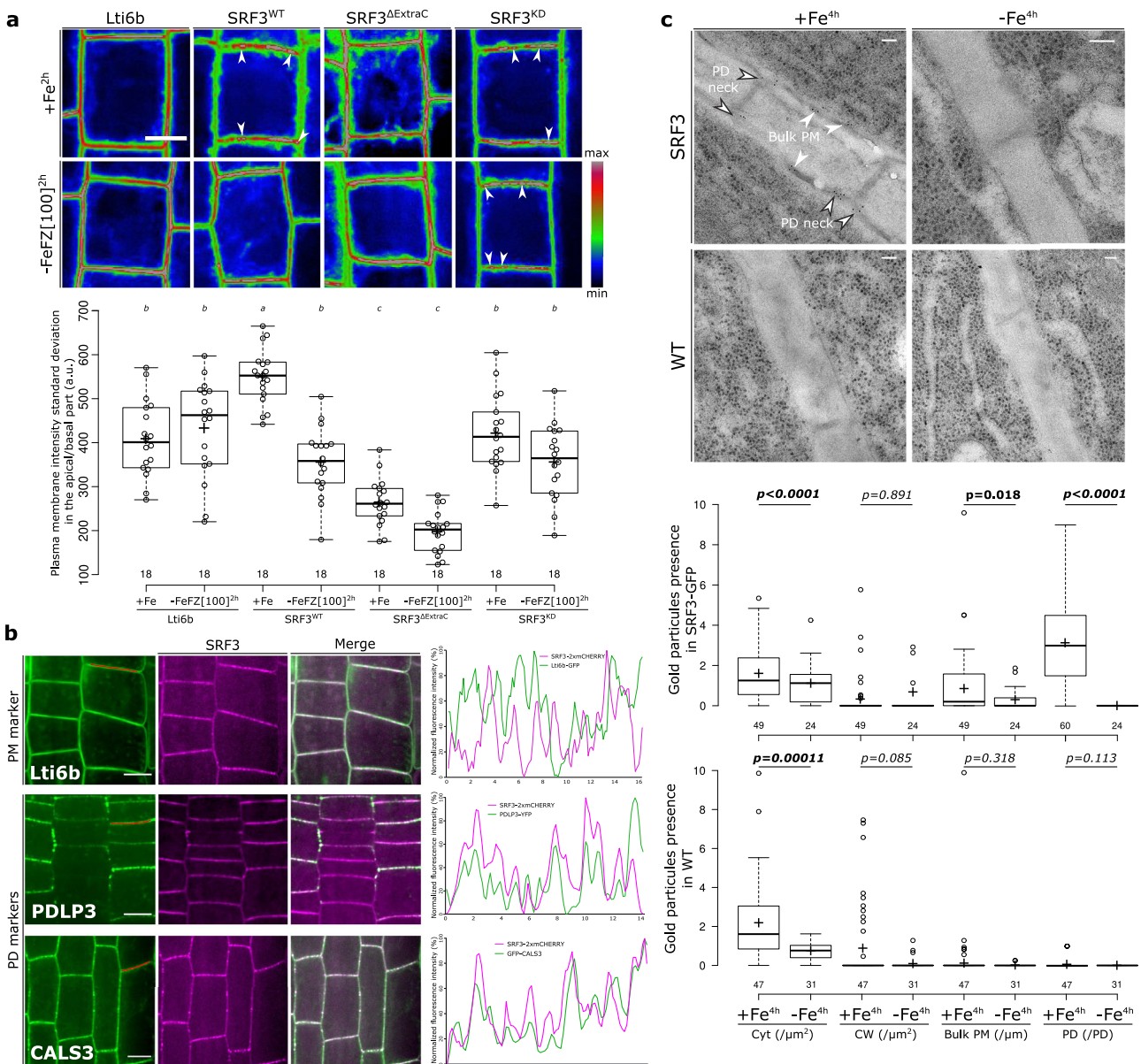

**Fig. 3 | SRF3 co-exists in two sub-populations at the plasma membrane, which decrease under low iron levels. a** Confocal images of root epidermal cells of 5 day old seedlings expressing *p35s::eGFP-Lti6b*, *pUBQ10::SRF3-mCITRINE*, *pUBQ10::SRF3WT-mCITRINE*, *pUBQ10::SRF3ΔExtraC-mCITRINE*, *pUBQ10::SRF3KD-mCITRINE* under iron sufficient (+Fe, 2 h) or low iron (−FeFZ 100 μM, 2 h) conditions and the related quantification [ANOVA with post hoc Tukey test; Letters: statistical differences ($p < 0.05$)]. Note that the pictures have been pseudo-colored in rainbow RGB and intensities are optimized to emphasize changes in plasma membrane polarity and localization in the punctuated foci. Arrows indicate punctuated structures. Scale bars, 10 μm. **b** Confocal images of root epidermal cells in the transition-elongation of 5 day old seedlings co-expressing, *p35s::eGFP-Lti6b*, *pPDLP3-PDLP3-YFP, 3Ss::CALS3-GFP*, left, with *pUBQ10::SRF3-2xmCHERRY*, middle and the relative merge of the two pictures. Red line on the left image indicates where the scan line has been traced. Right panel: graphs showing the signal intensity in both channel on the apical-basal part of the cell along the scan line. This co-localization has been observed on three independent experiments on at least 4 roots. Scale bars, 10 μm. **c** Electron micrograph of gold particle detected with anti-GFP antibody in plants expressing *pSRF3::SRF3-GFP* (SRF3) and the related control in *Ler-0* background under sufficient (+Fe, 4 h) and low (−Fe, 4 h) iron media and the related quantification [two-ways Mann-Whitney test coupled with Montecarlo correction ($p = 0.05$)]. Cyt, cytosol; CW, cell wall; PM plasma membrane; PD plasmodesmata. Scale bars, 50 nm.

activate *IRT1* transcription by inhibiting callose synthases with DDG. The addition of DDG that does not impact root iron content (Supplementary Fig. 8a), strongly reduced *IRT1* promoter activation in WT and was not observed in the *srf3-4* mutant compared to low iron only (Fig. 5b). Altogether, these observations indicate that *SRF3*-dependent callose synthases activity ultimately tunes the expression of the major root iron transporter *IRT1*.

We then investigated whether SRF3 and callose synthases can co-exist in the same region of the plasma membrane in roots. For this, we employed co-localization analysis using dual-color total internal reflection fluorescence (TIRF). These experiments revealed that SRF3 was organized in domains in the bulk PM fraction that partially co-localized with CalS3 but not with the β−1,3-glucanases reporter PdBG1 known to negatively regulate callose deposition[26] (Fig. 5c). This suggested a possible regulatory role of SRF3 on callose synthases rather than on β−1,3-glucanases. To assess this hypothesis, we crossed *SRF3-OX*, which does not present any root growth defects (Fig. 5d), with a mutated version of *CalS3* (*cals3-3d*) whose activity is up to 50% higher

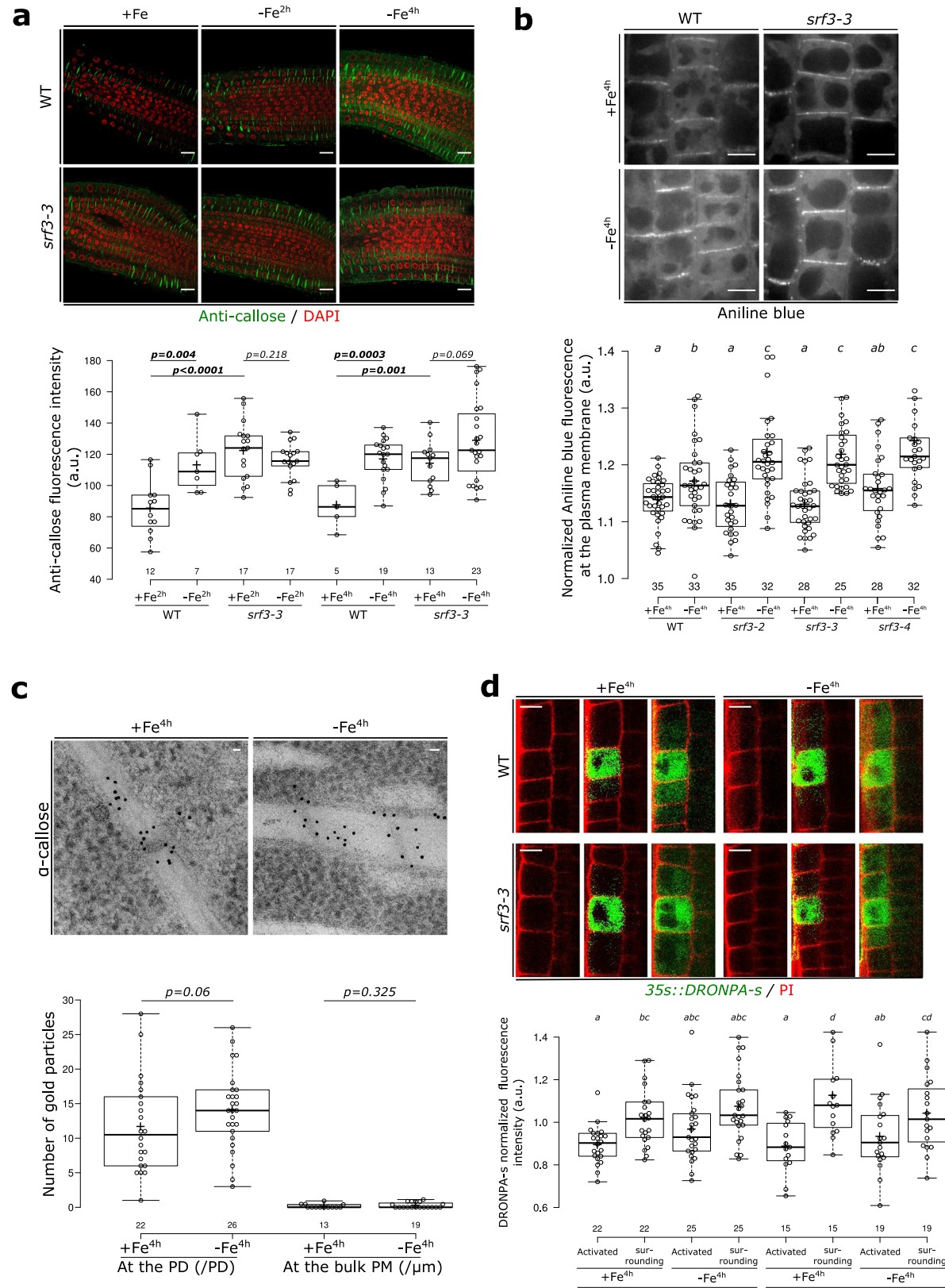

and subsequently accumulates more callose, resulting in shorter roots than in WT (Fig. 5d)[24]. Surprisingly, the double homozygous lines of *SRF3-OX/cals3-3d* showed a further decrease of root growth compared to the *cals3-3d* single mutant (Fig. 5d). Since ectopic expression of *SRF3* did not suppress the *cals3-3d* root growth phenotype, these data support a model in which *SRF3* acts together with *Cals3* to regulate

root growth. However, this phenomenon seems to be independent of callose synthases-mediated callose biosynthesis because *SRF3* over-expression, acting as a negative regulator or callose deposition, was not able to recover callose overaccumulation-related *cals3-3d* root growth phenotype. To test the independence of callose biosynthesis, we crossed *SRF3-OX* with *35s::GFP-PDLP5* (*PDLP5-OX*) line. *PDLP5-OX*

**Fig. 4 | SRF3 is a negative regulator of callose deposition but does not regulate cell-to-cell movement of proteins. a** Confocal images of root meristems of 5 day old seedlings stained with callose antibody (green) and DAPI to stain the nucleus (red) under sufficient (+Fe, 2 h and 4 h) and low (−Fe, 2 h and 4 h) iron media and the related quantification [two-ways student test ($p < 0.05$), n.s.: non-significant]. Scale bars, 10 μm. **b** Confocal images of root epidermal cells in the elongation zone of 5 day old seedling stained with aniline blue under sufficient (+Fe, 4 h) and low (−Fe, 4 h) iron media in WT and *srf3-3* and the related quantification [ANOVA with post hoc Fisher test; Letters: statistical differences ($p < 0.05$)]. Scale bars, 10 μm. **c** Electron micrograph of gold particle detected with anti-callose antibody in plants expressing *pSRF3::SRF3-GFP* (SRF3) grown on sufficient (+Fe, 4 h) and low (−Fe, 4 h) iron media and the related quantification [one-way student test ($p < 0.1$), n.s.: non-significant]. PD, plasmodesmata; PM, Plasma membrane. Scale bars, 50 nm. **d** Confocal images of root epidermal cells in the transition-elongation zone of 5 day old seedlings expressing *p3Ss::DRONPA-s* in WT and *srf3-3* under sufficient (+Fe) and low (−Fe) iron media and the related quantification [ANOVA with post hoc Fisher test; Letters: statistical differences ($p < 0.05$)]. Scale bars, 10 μm.

displays root growth and callose deposition phenotypes similar to *cals3-3d* as PDLP5 most likely acts as a positive regulator of *CalS1 and CalS8*-mediated callose biosynthesis[28,29]. For this cross the root growth was indistinguishable from the *PDLP5-OX* line, supporting the model that *SRF3* might control root growth with *CalS3* independently of its callose biosynthesis activity (Supplementary Fig. 8b). Finally, to test the involvement of CalS3/SRF3 axis in regulating root growth under low iron conditions, we obtained the *cals3-6* mutant. This line displays neither root growth nor callose accumulation defects in standard growth conditions[24]. *cals3-6* was more sensitive to low iron compared to the WT, thus mimicking the phenotype of the *SRF3-OX* line and therefore further indicating that *CalS3* is required to modulate root growth under low iron (Fig. 5e). Surprisingly, in the double mutant *cals3-6/SRF3-OX*, root growth was less sensitive to low iron conditions compared to WT, supporting that these two genes regulate root growth under low iron conditions (Fig. 5e). As no additive hypersensitivity was observed (Fig. 5e), these data suggested that they might be involved in the same pathway. To further test this, we made use of the *CALS3-OX* line, which is less sensitive to low iron and thus presents the opposite response compared to *SRF3-OX* in reference to WT (Fig. 5f). Using these lines, we generated the double mutant *CalS3-OX/SRF3-OX*, which behaved like *CalS3-OX* compared to WT, indicating that *CalS3* and *SRF3* are both involved in the same pathway and that *SRF3* might act downstream of *CalS3* (Fig. 5f). We then went on to further characterize the relation of *SRF3* and *CalSs*. We reasoned that if CalSs were upstream of SRF3, the inhibition of callose synthase activity would impact SRF3 PM levels. However, co-treatment of low iron with DDG did not modify PM-associated SRF3 levels and therefore suggested that callose synthase is downstream of SRF3 (Supplementary Fig. 8c). Finally, monitoring the early and late root growth rate of WT and *srf3-3* during the application of DDG and low iron levels showed a partial complementation of *srf3-3* root growth phenotype thus further supporting a downstream role of *CalSs* with regards to *SRF3* (Fig. 5g and Supplementary Fig. 8d). In sum, our data suggest that *SRF3* acts along with callose synthases early-on upon low iron levels to regulate iron homeostasis and root growth, however, we were not able to directly decipher the sequentiality of the pathway hinting towards a complex functional interaction of *SRF3* and *CalSs*.

## SRF3 coordinates iron homeostasis and bacteria elicited immune responses

*SRF3* was originally identified as a genetic locus underlying immune-related hybrid incompatibility in *Arabidopsis* and shown to be involved in bacterial defense-related pathways in leaves[30]. Gene ontology (GO) analysis of root RNAseq data in iron sufficient condition and the analysis of the root specific *pCYP71A12::GUS* immune reporter upon treatment with the bacterial elicitor flg22 showed that SRF3 is acting in bacterially elicited root immunity responses (Fig. 6a and Supplementary Fig. 9a). We then investigated the specificity of SRF3's role by assessing the late root growth responses to different pathogen-associated molecular patterns (PAMPs) and plant-derived damage-associated molecular patterns (DAMPs). *srf3* roots were only impaired in their response to flg22 but not to chitin or pep-1 compared to WT (Supplementary Fig. 9b, c). Similar to the low iron levels response, the

flg22 root growth response was already apparent early-on in WT and absent in *srf3* mutants (Fig. 6b, Supplementary Fig. 9d, e, and Movies 8, 9). To test whether the increased iron content of *srf3* might be related to this response, we analyzed the early and late root growth responses to flg22 of *bts-1* and *opt3-2*. Both of these mutants responded like WT, indicating that higher iron content in the root does not generally affect root growth upon immune response elicitation[13,14] (Supplementary Fig. 9f–h). We therefore concluded that the role of *SRF3* in controlling early root growth upon bacterial elicitation is specific and most likely related to its signaling activity.

Because of the similar growth response to low iron levels and flg22 treatment, we hypothesized that the SRF3-dependent root growth regulation to these two cues might rely on a similar molecular mechanism. Consistent with this idea, we found that upon flg22 addition, the SRF3 protein displays similar cellular dynamics as observed under low iron condition (Fig. 6c and Supplementary Figs. 5c and 9i) while no significant changes of SRF3 transcriptional regulation were observed (Supplementary Fig. 9j). Therefore, SRF3 appears to be a point of convergence between iron and flg22-dependent signaling mediating root growth regulation.

One model explaining this convergence could be that flg22 might trigger a transient decrease of cellular iron levels, thus promoting SRF3 degradation. We therefore performed RhoNox-1 staining after 1 h of flg22 treatment and observed a decrease of fluorescence compared to mock treatment in the elongation zone (Fig. 6d). This indicates a swift and local decrease of iron concentration in roots upon flg22 stimulus, which is consistent with flg22-triggered rapid *IRT1* expression (Supplementary Fig. 10a and Supplementary Movie 10). Moreover, mining of publicly available root RNAseq data revealed a broad impact of short-term flg22 treatment on the expression of iron homeostasis genes (Supplementary Data S3)[31]. We then wondered whether the flg22-triggered iron deficiency responses relied on SRF3-dependent callose synthase activity. Co-treatment of WT roots with DDG and flg22 led to a decrease of *IRT1* promoter activity compared to flg22 treatment alone. In *srf3-4* a slight increase of *IRT1* promoter activity under flg22 was observed compared to mock condition but to a lesser extent than the WT (Fig. 6e). However, it remained insensitive upon co-treatment compared to WT (Fig. 6e). Overall, these data indicate that flg22-dependent *IRT1* activation relies on SRF3-mediated callose synthase activity as observed under low iron conditions.

Finally, to investigate the extent of *SRF3*-dependent coordination of bacterial immune responses and iron homeostasis, we performed a RNAseq analysis after 2 h of exposure to low iron levels or flg22 in *srf3* mutants and WT roots. Strikingly, in WT, 90% of the differentially expressed genes (DEGs) in low iron overlapped with DEGs upon flg22 treatment and were regulated in the same manner (e.g., upregulated DEGs in low iron were also upregulated upon flg22). Importantly, these DEGs were not associated with a general stress response since only few of them were overlapping with those in cold, heat, drought, salt and mannitol datasets[32,33] (Fig. 6f and Supplementary Fig. 10b, Data 4 and 5). To further confirm that low iron levels trigger immunity genes and to validate the RNAseq, we conducted qPCR for two early markers of flg22-triggered immunity, *FRK1* and *MYB51* which showed a transient activation of the two genes within 4 h (Supplementary Fig. 10c)[34]. To

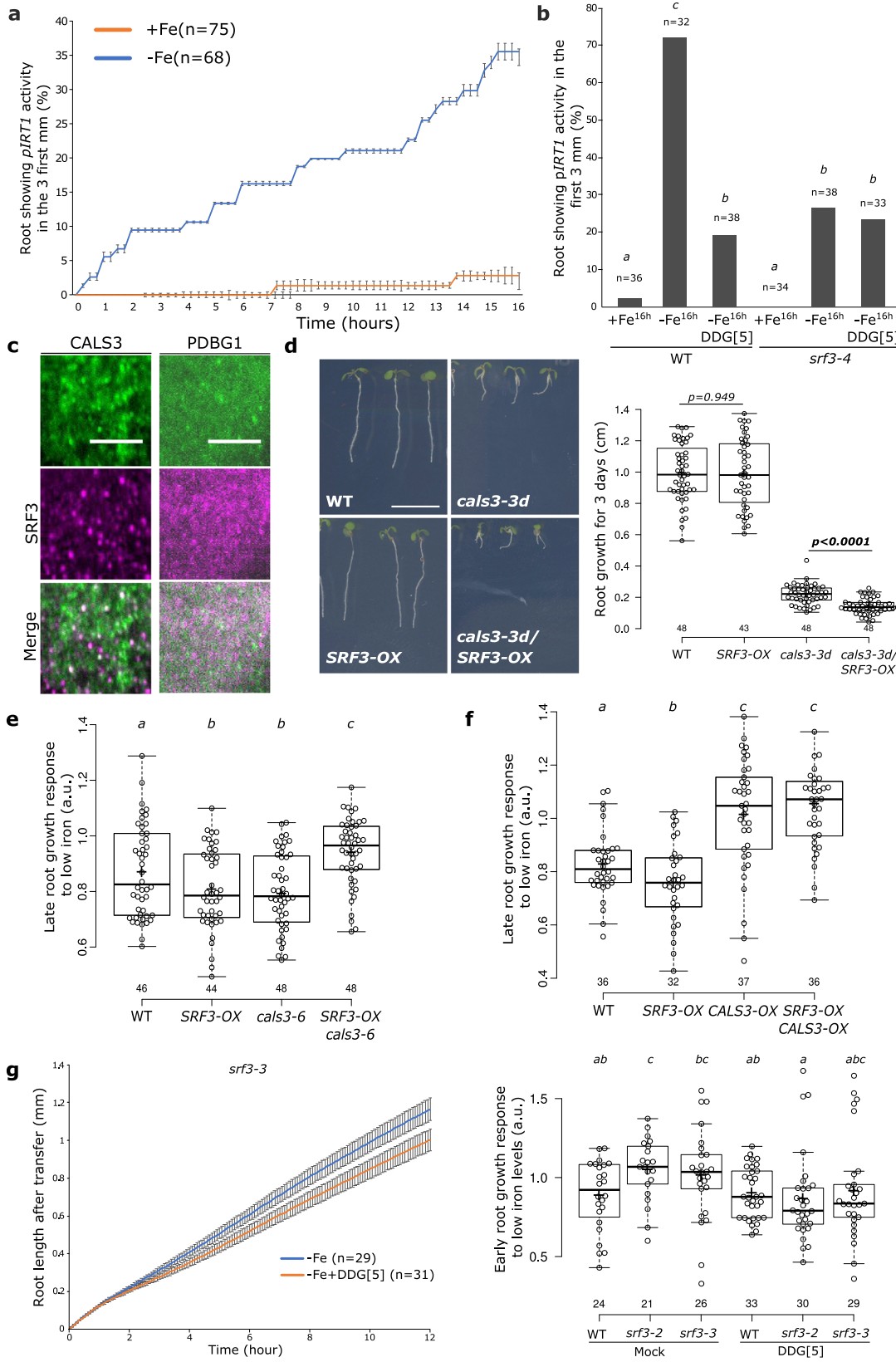

determine how much of this common transcriptional program is coordinated by *SRF3*, we analyzed the *srf3* transcriptome datasets. DEGs in flg22 and low iron in *srf3* mutants only overlapped by 24% demonstrating that *SRF3* coordinates a large part of the common transcriptional program that is triggered in response to early response to low iron and flg22 (Fig. 6f). Overall, our work establishes SRF3 as a

major coordinator of bacterial immune response and iron deficiency signaling pathways, which relies on callose synthase activity.

## Discussion

Guided by a GWAS peak, we have identified an LRR-RK, *SRF3* as an integrator of multiple signaling pathways upon perception of low iron

**Fig. 5 | Regulation of *IRT1* and root growth by SRF3-dependent callose synthase activity under low iron levels. a** Graph representing the quantification of *pIRT1::NLS-2xYPet* time-lapse analysis under mock (+Fe) and low iron levels (−Fe) [error bars indicate Standard error of the mean (SEM)]. **b** Graph representing the percentage of root showing *IRT1* promotor activation under sufficient (+Fe, 16 h) and low (−Fe, 16 h) iron media in WT and *srf3-4* in presence or absence of DDG. [ANOVA with post hoc Tukey test; Letters: statistical differences ($p < 0.05$)]. **c** Micrographs of 5 day old seedlings taken in the root elongation zone of plants expressing *35s::GFP-CALS3* (upper) and *UBQ10::SRF3-2xmCHERRY* (middle) and merge channel (lower) acquired by TIRF microscopy. This pattern has been observed on at least 15 roots throughout 2 independent experiments. Scale bars, 5 μm. **d** Picture of 9 day old seedlings of WT, *cals3-3d*, *pUBQ10::SRF3-mCITRINE* (*SRF3-OX*) and *cals3-3d/SRF3-OX* and the related quantification [two-ways student test ($p < 0.05$), n.s. non-significant]. Scale bar, 1 cm. **e**, **f** Boxplots of late root growth response of plants grown for 3 days in low iron levels. [ANOVA with post hoc Fisher (**e**) and Tukey tests (**f**); Letters: statistical differences ($p < 0.05$). **g** Graph showing time-lapse of the root length for 12 h of *srf3-3* under low iron (−Fe) media in presence or absence of DDG and the related quantification including the *srf3-2* mutant [ANOVA with post hoc Tukey test; Letters: statistical differences ($p < 0.05$); Error bars indicate SEM].

conditions. Our data is consistent with a hypothetical model in which SRF3 is involved in sensing and transducing the early lack of iron, which elicits CalSs signaling in the root tip to modulate growth, iron homeostasis, and bacterially elicited immune responses by a yet unknown mechanism. As this is highly reminiscent of nutritional immunity conferred by the TfR mammalian and *Drosophila* systems that sense iron levels and control iron and immune responses, we propose that *SRF3* is important in mediating plant nutritional immunity[3,5].

We discovered that root growth is modulated within the first 4 h upon exposure to low iron levels looking at earlier time points than usually considered[13,21,35,36] (Fig. 2a). This early response is SRF3-dependent, exposing this LRR-RK as being a key part of the genetically encoded ability of roots to perceive and transduce low environmental iron levels. In light of other LRR-RK signaling transduction mechanisms[17], our results lead towards the following model for SRF3 (1) the LRR extracellular domains senses a signal that is informative of the early lack of iron, (2) which triggers decrease of its levels at the PM, (3) to regulate early root growth. Based on our RNAseq analysis, we also found that the role of SRF3 in transducing low iron levels at an early stage is not restricted to the root growth regulation (Figs. 1e, 5b, and 6). However, we did not provide direct evidence of the involvement of SRF3 signal transduction to regulate iron homeostasis and bacterial immune pathways. This is very likely since SRF3 is known to be part of the phosphorelay upon PAMP immune response[37]. Altogether, our data indicate that roots perceive external variation of iron rapidly through SRF3-dependent signal transduction to coordinate root signaling pathways.

We have found that SRF3 acts on iron-induced callose synthases to mediate proper signaling. Surprisingly, cell-to-cell movement of proteins were not affected early-on upon low iron levels, despite callose synthases activation and increased callose deposition (Fig. 4c, d). However, it is possible that callose deposition might impact later responses since callose deposition-mediated plasmodesmata closure can take hours to days to occur[38] or has a different function early-on as recently reported in leaves[39]. Moreover, the observed increase of callose deposition, tuning of iron homeostasis and activation of immune responsive genes might be a defense priming mechanism along the lines of the recently described roles of siderophores[40,41]. Thus, SRF3-dependent regulation of root growth, iron homeostasis and defense signaling pathways does not rely on impeding cell-to-cell movement thereby putting the spotlight onto a likely early signaling function of callose synthases independent of their biosynthetic activity. Moreover, further investigations need to be conducted to decipher the direct implication of *CalS3* and other *CalSs* in this pathway. Taken together, we propose that a biosynthesis independent function of callose synthases, which is SRF3 dependent, is required to regulate early root growth, iron homeostasis and defense signaling pathways under low iron levels.

Our work highlights that *SRF3*-mediated signaling is at the nexus of the early root responses to low iron and bacterial-derived signals. RNAseq analysis revealed that early responses to low iron and flg22 are highly similar and largely coordinated by *SRF3* (Fig. 6f and

Supplementary Data 2, 4). The early lack of iron can activate the PTI signaling pathways in a *SRF3*-dependent manner (Fig. 6f and Supplementary Fig. 9c). This suggests that SRF3 is perceiving either environmental low iron levels or flg22-mediated iron decrease (Fig. 6d) and might interact with signal transduction components involved in PTI. Further supporting this hypothesis, we found that root growth response of *fls2* mutants, which are impaired in PTI-triggering immunity, is decreased under low iron levels (Supplementary Fig. 10d–h). During the early perception of flg22 in leaves, different *SRF3* natural alleles tune the phosphorylation status of MAPKs that are acting downstream of FLS2[30]. In roots, we could not detect any phosphorylation defects when exposed to flg22 for 5 and 10 min, suggesting that SRF3 might be involved in the BOTRYTIS-INDUCED KINASE 1 (BIK1) and RBOH pathways rather than MAPKs signaling pathways (Supplementary Fig. 10d, i). Altogether, our observations lead to a model in which SRF3 perceives an early lack of iron or flagellin-derived signals to modulate iron homeostasis and PTI signaling pathways through the regulation of BIK1-RBOH pathways.

During host–pathogen interactions, withholding iron to limit pathogen virulence is an early host line of defense, which is part of the nutritional immune responses as previously reported in vertebrates and invertebrates[4,5]. Eliciting bacterial immune responses triggers a SRF3-dependent decrease of cellular iron levels, showing a conserved principle of this nutritional immune response being present in plants (Figs. 1f and 6d). In line with this idea, we have found that the lack of *SRF3* impedes mechanisms related to the ability of root tissues to withhold iron such as those conferred by ZIF1 and NAS4[42,43] (Fig. 1d and Supplementary Data 2). Moreover, similar to the nutritional immunity systems described in mammals and *Drosophila melanogaster* based on TfR, SRF3 senses the immediate lack of iron, which is also relayed to a common signaling pathway linking low iron and immunity responses[3] (e.g., BMPR; Fig. 6f). Altogether, we therefore propose that SRF3 is a central player in a mechanism that embodies a fundamental principle of nutritional immunity by coordinating bacterial immunity and iron signaling pathways via sensing iron levels.

## Methods
### Plant materials and growth conditions
For surface sterilization, *Arabidopsis thaliana* seeds of 231 accessions from the Regmap panel (Supplementary Data 1) that had been produced under uniform growth conditions were placed for 1 h in opened 1.5-mL Eppendorf tubes in a sealed box containing chlorine gas generated from 130 mL of 10% sodium hypochlorite and 3.5 mL of 37% hydrochloric acid. For stratification, seeds were imbibed in water and stratified in the dark at 4 °C for 3 days. Seeds were then put on the surface of 1x MS agar plates, pH 5.7, containing 1% (w/v) sucrose and 0.8% (w/v) agar (Duchefa Biochemie) using 12 × 12-cm square plates. The iron-sufficient medium contained 100 μM Na-Fe-EDTA and the iron-deficient (1xMS iron free) medium contained 300 μM Ferrozine, a strong iron chelator [3-(2-pyridyl)−5,6-diphenyl-1,2,4-triazinesulfonate, Sigma-Aldrich]. This condition was only used for GWAS. For further experimentation, we used the Fe -sufficient or -free media described in Gruber et al., with no or a decrease level of Ferrozine, 100,

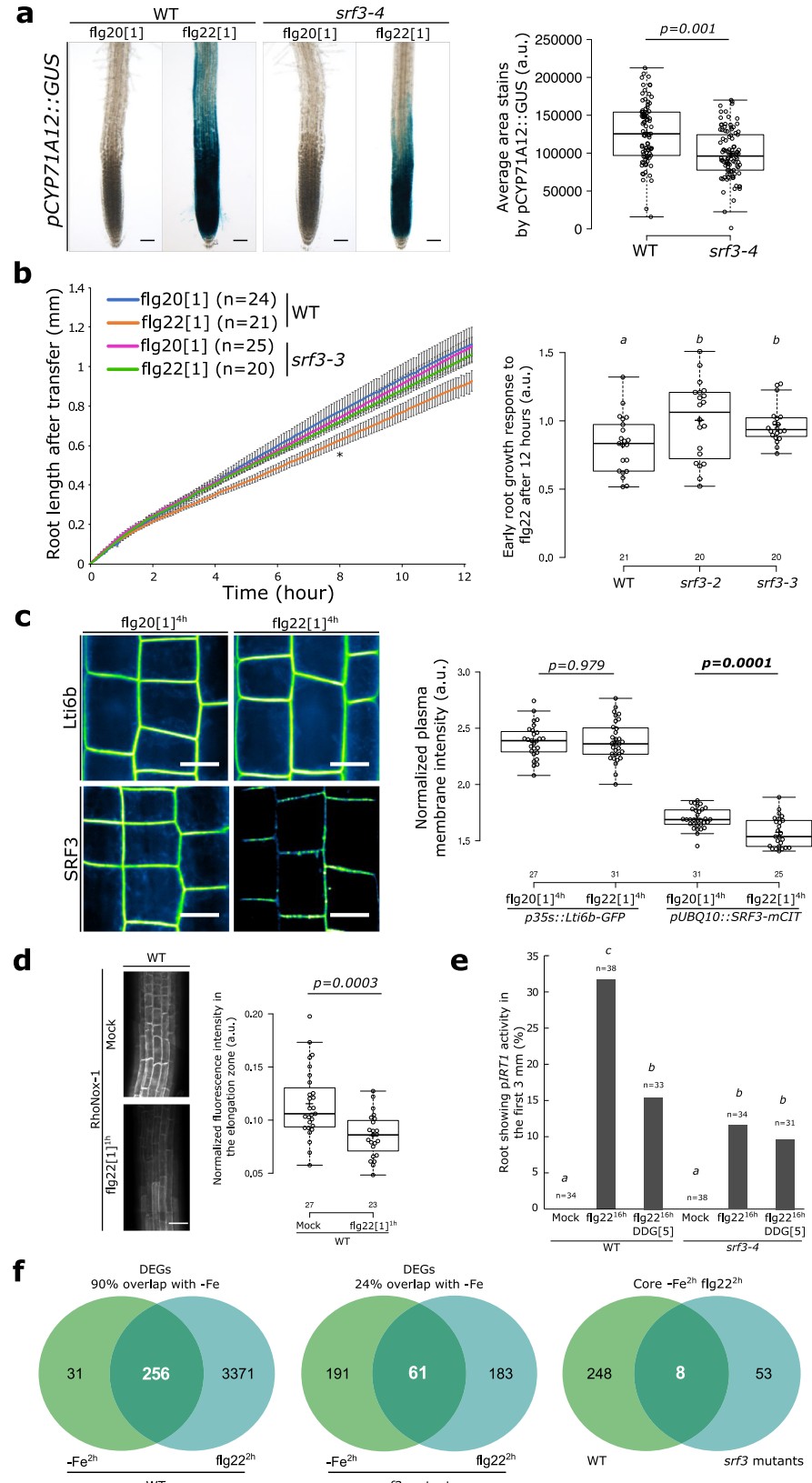

50, and 10 μM[58]. Using the Gruber et al. iron-free medium, we add 300 μM of Na-Fe-EDTA to test *srf3* phenotype under iron excess[44]. The Gruber et al. media contains, 750 μM of $MgSO_4-7H_2O$, 625 μM of $KH_2PO_4$, 1000 μM of $NH_4NO_3$, 9400 μM of $KNO_3$, 1500 μM of $CaCL_2-2H_2O$, 0.055 μM of $CoCL_2-6H_2O$, 0.053 μM of $CuCL_2-2H_2O$, 50 μM of $H_3BO_3$, 2.5 μM of KI, 50 μM of $MnCl_2-4H_2O$, 0.52 μM of

$Na_2MoO_4-2H_2O$, 15 μM of $ZnCL_2$, 75 μM of Na-Fe-EDTA (sufficient media) or 0 μM of Na-Fe-EDTA (low iron media), 1000 μM of MES adjusted to pH 5.5 with KOH. The *srf3-2, srf3-3, bts-1, opt3-2, fls2-c, fls2-9, vit-1, and cals3-3d* mutant lines are in Col-0 background and were described and characterized[13–15,24,30,45,46]. The reporter lines, *35Ss::PdBG1-mCITRINE, 35 s::GFP-CALS3, pPDLP3::PDLP3-YFP, 35s::GFP-*

**Fig. 6 | Coordination of bacterial immunity and iron homeostasis signaling pathways by SRF3. a** Representative pictures of roots expressing *pCYP71A12::GUS* in WT and *srf3-4* under flg20 and flg22 treatment (1 μM, 24 h) and the related quantification [two-ways student test ($p < 0.05$)]. Scale bars, 50μm. **b** Graph showing time-lapse analysis of root length of WT and *srf3-3* upon flg20 and flg22 treatments (1 μM) [error bars indicates standard error of the mean (SEM); Asterix: first timepoint with significant difference between +Fe and −Fe for the WT according to a mixed effect model ($p < 0.05$)] and the related quantification including the *srf3-2* mutant [ANOVA with post hoc Tukey test; Letters: statistical differences ($p < 0.05$)]. **c** Confocal images of root epidermal cells of 5-day-old seedling expressing *p35S::eGFP-Lti6b* and *pUBQ10::SRF3-mCITRINE* in flg20 and flg22 (1 μM, 4 h) and the related quantification [two-ways student test ($p < 0.05$); n.s.: non-significant]. Scale bars, 10 μm. **d** Confocal images of root epidermal cells of 5 day old seedlings stained with RhoNox-1 in WT in mock or flg22 (1 μM, 1 h) and the related quantification in the elongation zone [two-ways student test ($p < 0.05$)]. Scale bars, 50 μm. **e** Graph representing the percentage of root showing *IRT1* promotor activation under mock and flg22 (1 μM, 16 h) treatment in WT and *srf3-4* in presence or absence of DDG. [ANOVA with post hoc Tukey test; Letters: statistical differences ($p < 0.05$)]. **f** Venn diagram of differentially expressed genes under low iron levels (−Fe, 2 h) and flg22 (1 μM, 2 h) in WT (left) in *srf3* (middle) and DEGs in both condition between WT and *srf3*.

*PDLP5, 35s::eGFP-Lti6b 35 s::DRONPA-s, pSUC2::GFP* and *pCYP71A12::GUS* are in Col-0 background and were described and characterized[24,26,47–50]. The T-DNA insertion lines for SRF3, SAIL1176_B01 (*srf3-4*) and SALK_202843, as well as for *at4g03400*, SAIL_811_C06 (*at4g03400*) were purchased from Nottingham Arabidopsis Stock Center (NASC, Nottingham, United Kingdom). The primers used for genotyping the T-DNA lines are listed below (List of primers, Supplementary Data 6). Plants were grown in long day conditions (16/8 h) in walk in growth chambers at 21 °C, 50 μM light intensity, 60% humidity. During night-time, temperature was decreased to 15 °C.

## Plant transformation and selection
Each construct (see below: "Construction of plant transformation vectors (destination vectors) and plant transformation"), was transformed into C58 GV3101 *Agrobacterium tumefaciens* strain and selected on YEB media (5 g/L beef extract; 1 g/L yeast extract; 5 g/L peptone; 5 g/L sucrose; 15 g/L bactoagar; pH 7.2) supplemented with antibiotics (Spectinomycin, Gentamycin). After 2 days of growth at 28 °C, bacteria were collected using a single-use cell scraper, resuspended in about 200 mL of transformation buffer (10 mM MgCl$_2$; 5% sucrose; 0.25% silwet) and plants were transformed by the floral dipping method[51]. Plants from the Columbia-0 (Col-0) accession were used for transformation. Primary transformants (T1) were selected in vitro on the appropriate antibiotic/herbicide (glufosinate for mCITRINE, hygromycin for mCHERRY and mSCARLET tagged proteins). Approximately 20 independent T1s were selected for each line. In the T2 generation at least three independent transgenic lines were selected using the following criteria when possible: (i) good expression level in the root for detection by confocal microscopy, (ii) uniform expression pattern, (iii) single insertion line (1 sensitive to 3 resistant segregation ratio) and, (iv) line with no obvious abnormal developmental phenotypes. Lines were rescreened in T3 using similar criteria as in T2 with the exception that we selected homozygous lines (100% resistant). At this step, we selected one transgenic line for each marker that was used for further analyses and crosses.

## GWA mapping
Analogous to previous successful GWAS studies using 200–300 diverse Arabidopsis accessions[52–56], 231 diverse natural accessions (12 plants/accession were planted) were grown on 1× MS agar plates containing 300 μM Ferrozine under long day conditions (16 h light) at 21 °C. Plant images were acquired by EPSON flatbed scanners (Perfection V600 Photo, Seiko Epson CO., Nagano, Japan) every 24 h for 5 days (2 DAG–6 DAG). Root image analyses and trait quantification were performed using the BRAT software[57]. Median root growth rate ($n \geq 3$) values between 4 to 5 days were used for GWA study. For more accuracy, the roots not detected or not germinated were not included in the analyses. Population structure corrected GWA mapping was conducted using AMM with the 250 K SNP dataset (by using the GWAS pipeline implemented on https://gwas.gmi.oeaw.ac.at/)[58]. For population structure control, this webservice uses an identity by state (IBS) genetic relatedness matrix, first a priori for the full genotype dataset and then by removing the contributions of SNPs of accessions that are not contained in the GWAS trait dataset. SNPs with minor allele counts equal or greater to 10 were taken into account. To control for false positives, we used a 5% FDR threshold calculated by the Benjamini–Hochberg–Yekutieli method to correct for multiple testing[59]. The GWAS peak in proximity of *SRF3* (Fig. 1a) contained 4 significant SNPs. By analyzing the unique combinations of these 4 SNPs in the 231 accessions, four groups of haplotypes were defined as Group A, Group B, Group C, and Group D.

## Phenotyping of early root growth responses
Seeds were sowed in +Fe media described in Gruber et al. and stratified for 2–3 days at 4 °C[58]. Five days after planting, about 15 seedlings were transferred to a culture chamber (Lab-Tek, Chamberes #1.0 Borosilicate Coverglass System, catalog number: 155361) filled with −Fe or +Fe medium described in Gruber et al. or +Fe medium containing flg20 or flg22[58]. Note that the transfer took about 45–60 s. Images were acquired every 5 min for 12 h representing 145 images per root in brightfield conditions using a Keyence microscope model BZ-X810 with a BZ NIKON Objective Lens (2x) CFI Plan-Apo Lambda.

## Phenotyping of late root growth responses
Seeds were sowed in +Fe media described in Gruber et al. and stratified for 2–3 days at 4 °C[58]. Five days after planting, 6 plants per genotype were transferred to four 12 × 12-cm plates in a pattern in which the positions of the genotypes were alternating in a block design (top left, top right, bottom left and bottom right). After transfer, the plates were scanned every 24 h for 3 days using the BRAT software[57].

## SRF3 transcript expression by RT-PCR
Total mRNA was extracted from wild type, and three insertions each of: *pUBQ10::SRF3-mCITRINE, pUBQ10::SRF3$^{KD}$-mCITRINE,* and *pUBQ10::SRF3$^{\Delta ExtraC}$-mCITRINE* using Sigma-Aldrich Spectrum Plant Total RNA Kit. cDNA was produced using Applied Biosystems (by Thermo Fisher Scientific) High-Capacity cDNA Reverse Transcription Kit. The expression of SRF3 and the ubiquitous TCTP transcripts was tested by PCR using primers starting with "RT" in Data S6.

## Quantitative real-time PCR
For *SRF3* expression analysis seedlings were grown initially on iron sufficient media (1x MS, 1% w/v Caisson Agar) for 5 days and then shifted to either iron sufficient or low iron (100 μM FerroZine) 1xMS liquid medium. Nylon mesh (Nitex Cat 03-100/44; Sefar) was placed on top of the solidified media to facilitate transfer. Root tissues were collected for RNA extraction 3 h post transfer by excision with fine scissors. Each biological replicate was constituted by RNA extraction from 30 to 40 whole roots. Samples were immediately frozen in liquid nitrogen, ground, and total RNA was extracted using the RNeasy Plant Mini kit (QIAGEN GmbH, Hilden, Germany). qRT-PCR reactions were prepared using 2x SensiMix SYBR & Fluorescein Kit (PEQLAB LLC, Wilmington, DE, USA) and PCR was conducted with a Roche Lightcycler 96 (Roche) instrument. Relative quantifications were performed for all genes with the β-tubulin gene (AT5G62690) used as an internal

reference. The primers used for qRT-PCR are shown in list of primers (Supplementary Data 6).

## RNAseq

Total RNA was extracted from roots of plants 5 days after germination using RNA protein purification kit (Macherey-Nagel). Next generation sequencing (NGS) libraries were generated using the TruSeq Stranded mRNA library prep kits (Illumina, San Diego, CA, USA). Libraries were sequenced on a HiSeq2500 (Illumina, San Diego, CA, USA) instrument as single read 50bases. NGS analysis was performed using Tophat2 for mapping reads onto the Arabidopsis genome (TAIR10)[60,61], HT-seq for counting reads and EdgeR for quantifying differential expression[62]. We set a threshold for differentially expressed genes (Fold change (FC) > 2 or FC < −2, FDR < 0.01). Gene ontology analysis was performed using the AgriGOv2 online tool[63]. Venn diagrams were generated with the VIB online tool (http://bioinformatics.psb.ugent.be/webtools/Venn/). The plot in Fig. 3a was generated using the online Revigo software (http://revigo.irb.hr/).

## Microscopy setup

All imaging experiments except when indicated below, were performed with the following spinning disk confocal microscope set up: inverted Zeiss microscope equipped with a spinning disk module (CSU-X1, Yokogawa, https://www.yokogawa.com) and the prime 95B Scientific CMOS camera (https://www.photometrics.com) using a 63x Plan-Apochromat objective (numerical aperture 1.4, oil immersion) or low-resolution 10x lens for time-lapse imaging. GFP, mCITRINE and RhoNox-1 staining were excited with a 488 nm laser (150 mW) and fluorescence emission was filtered by a 525/50 nm BrightLine® single-band bandpass filter. mSCARLET, mCHERRY and propidium iodide dyes were excited with a 561 nm laser (80 mW) and fluorescence emission was filtered by a 609/54 nm BrightLine® single-band band-pass filter (Semrock, http://www.semrock.com/). 405 nm laser was used to excite aniline blue and emission was recorded at 480–520 nm with 40x objectives. For quantitative imaging, pictures of root cells were taken with detector settings optimized for low background and no pixel saturation. Care was taken to use similar confocal settings when comparing fluorescence intensity or for quantification.

## FRAP experiment

Fluorescence in a rectangle ROI (50 μm², 15 μm long), in the plasma membrane region, was bleached in the root optical section by four successive scans at full laser power (150 W) using the FRAP module available on the Zeiss LSM 880 Airyscan 2. Fluorescence recovery was subsequently analyzed in the bleached ROIs and in controlled ROIs (rectangle with the same dimension in unbleached area). FRAP was recorded continuously during 90 s with a delay of 0.3 s between frames. Fluorescence intensity data were normalized using the equation: $I_n = [(I_t - I_{min})/(I_{max} - I_{min})] \times 100$ where $I_n$ is the normalized intensity, $I_t$ is the intensity at any time $t$, $I_{min}$ is the minimum intensity postphotobleaching, and $I_{max}$ is the mean intensity before photobleaching. For visualization, kymographs were obtained using kymograph function in Fiji.

## TIRF microscopy

Total internal reflection fluorescence (TIRF) microscopy was done using the inverted ONI Nanoimager from Oxford microscope with 100x Plan-Apochromat objective (numerical aperture 1.50, oil immersion). The optimum critical angle was determined as giving the best signal-to-noise ratio. Images were acquired with about 5–15% excitation (1 W laser power) and taking images every 100 ms for 500-time steps.

## DRONPA-s bleaching and activation

5 day old seedlings were transferred to a culture chamber (Lab-Tek, Chamberes #1.0 Borosilicate Coverglass System, catalog number: 155361) filled with −Fe or +Fe medium described in Gruber et al.[44] for 4 h. After 4 h, the cell wall was counter-stained by placing one drop of propidium iodide 15 μM (10 μg/mL in distilled water) on the root tip for 1 min. A coverslip was placed on the surface of the root for further imaging. DRONPA-s was bleached using the full laser power (150 W) of the 488 nm laser for 10 s. Then 2–4 regions of interest (ROIs) were drawn on the external lateral side of the epidermal root cells and DRONPA-s was activated in this region using the 405 nm laser doing 8 cycles at 15 W using the FRAP module available on the Zeiss LSM 880 Airyscan 2. Right after activation and then again 6 min later, images were acquired in both channel, PI and DRONPA-s.

## IRT1 reporter lines after 24 h

About 24 of 5 day old seedlings grown on iron sufficient medium were transferred to agar plate filled with +Fe or −Fe supplemented with 100 μM Ferrozine for 24 h. Fifteen seedlings were transferred to a culture chamber (Lab-Tek, Chamberes #1.0 Borosilicate Coverglass System, catalog number: 155361) filled with +Fe described in Gruber et al.[44]. The cell wall was counter-stained by placing one drop of propidium iodide 15 μM (10 μg/mL in distilled water) on the root tip for 1 min. A coverslip was placed on the surface of the root for further imaging. Images were acquired using the spinning disc set up described above using stitching and z-stack modes.

## Time-lapse imaging of IRT1 reporter lines

Five day old seedlings were grown on iron sufficient medium and then about 15 seedlings were transferred to a culture chamber (Lab-Tek, Chamberes #1.0 Borosilicate Coverglass System, catalog number: 155361) filled with +Fe or −Fe medium described in Gruber et al.[44] or +Fe medium containing flg22. Note that the transfer took about 45–60 s. Images were acquired every 20 min for 16 h representing 80 images per root using a Keyence microscope model BZ-X810 with BZ NIKON Objective Lens (2X) CFI Plan-Apo Lambda in brightfield, green (ET470/40x ET525/50 m T495lpxr-UF1) or red (ET560/40x ET630/75 m T585lpxr-UF1) channels.

## Time-lapse imaging of SRF3 transcriptional and translational reporter and control lines

About 15 5 day old seedlings grown on iron sufficient medium were transferred to a culture chamber (Lab-Tek, Chamberes #1.0 Borosilicate Coverglass System, catalog number: 155361) filled with +Fe or −Fe medium described in Gruber et al.[58]. Note that the transfer took about 45–60 s. Images were acquired every 10 min for 4 h using the spinning disc set up described above and assembled using the stitching mode, z-stack and definite focus options to keep track of the root and be localized at the same z-stage a long time, respectively.

## Cryofixation and freeze-substitution

Five day old seedlings of pSRF3::SRF3-GFP line (Landsberg erecta background) and Ler-0 were grown vertically on Caisson media complemented in iron. The seedlings were incubated for 4 h in liquid Caisson media, which were complemented or deficient in iron. Root tips were taken and cryofixed in 20% BSA filled copper platelets (100 nm deep and 1.5 mm wide) with EM PACT1 high-pressure freezer (Leica). The samples were transferred for freeze-substitution in AFS2 (Leica) at −90 °C in cryosubstitution mix: uranyl acetate 0.36%, in pure acetone, for 24 h. The temperature was raised stepwise by 3 °C per hour until reaching −50 °C and maintained for 3 h. The cryosubstitution mix was removed and replaced by pure acetone and then pure ethanol, for each of them three washes of 10 min were performed. The copper platelets were not removed in order to avoid sample loss. HM20 Lowicryl resin (Electron Microscopy Science) solutions of increasing concentrations were used for infiltration: 25% and 50% (1 h each), 75% (2 h), 100% (overnight, 4 h, 48 h-each bath was performed with new resin). The samples were then polymerized under ultraviolet

light for 24 h at −50 °C before raising the temperature stepwise by 3 °C per hour until reaching 20 °C and maintained for 6 h.

## Immunogold labeling

The samples were recovered by removing exceeding resin on the top and edges of the copper platelets. The latter were removed by applying alternatively heat shocks with liquid nitrogen and on a 40 °C heated knife to dissociate copper platelet from the resin. Ultrathin sections of 90 nm thickness were trimmed at a speed of 1 mm/s (EM UC7 ultra-microtome, Leica) and recovered on electron microscopy grids (T 300mesh cupper grids, Electron Microscopy Science) covered by 2% parlodion film. Once the grids were dry immunogold labeling was performed. The grids were successively incubated in 10 µl droplets of different reagents (0.22 µm filtered). The grids were first incubated in PHEM Tween 0.2% BSA 1% buffer (pH 6.9) for 1 min of rinsing before 30 min of blocking. The primary antibody anti-GFP rabbit polyclonal antibody (A11122, Thermo Fisher Scientific) and secondary antibody 10 nm colloidal gold-conjugated goat anti-rabbit IgG (Tebu-Bio) were diluted in PHEM Tween 0.2% BSA 1% buffer (pH 6.9) to 1:200 and 1:40, respectively, and grids were incubated for 1 h. Three rinsing steps of 5 min each were performed between the primary and secondary antibody incubation and after the secondary incubation. The grids were rinsed on filtered miliQ water droplets before drying and imaging. Image acquisition was performed at x42,000 magnification on a FEI Tecnai G2 Spirit TWIN TEM with axial Eagle 4 K camera.

## Immunolocalization of callose

Arabidopsis seedlings were grown on ½X MS 1% sucrose agar plate for 6 days and then incubated for 3 h in ½X MS 1% sucrose liquid medium for control condition or ½X MS 1% sucrose liquid medium containing 0.4 M mannitol, prior to fixation. The immunolocalization procedure was done according to Boutté et al.[64]. The callose antibody (Australia Biosupplies) was diluted to 1:300 in MTSB (Microtubule Stabilizing Buffer) containing 5% of neutral donkey serum. The secondary anti-mouse antibody coupled to TRITC (tetramethylrhodamine) was diluted to 1:300 in MTSB buffer containing 5% of neutral donkey serum. The nuclei were stained using DAPI (4',6-diamidino-2-phénylindole) diluted to 1:200 in MTSB buffer for 20 min. Samples were then imaged with a Zeiss LSM 880 using 40x oil lens. DAPI excitation was performed using 0.5% of 405 laser power and fluorescence collected at 420–480 nm; GFP excitation was performed using 5% of 488 nm laser power and fluorescence emission collected at 505–550 nm; TRITC excitation was performed with 5% of 561 nm power and fluorescence collected at 569–590 nm. All the parameters were kept the same between experiments to allow quantifications.

## Protein extraction and western blot for MAPK activity analysis

Protein extraction was carried out using 1 g of roots from 12 day old pUBQ10::SRF3-mCITRINE seedlings. Tissues were ground in liquid nitrogen and resuspended in 2 mL of ice-cold sucrose buffer (20 mM Tris, pH 8; 0.33 M Sucrose; 1 mM EDTA, pH 8; protease inhibitor). Samples were centrifuged for 10 min at $5000 \times g$ at 4 °C to remove cellular debris. Total proteins contained in the supernatant were centrifuged at 4 °C for 45 min at $20,000 \times g$ to pellet microsomes. The microsome pellet was resuspended in 1 mL of immunoprecipitation buffer (50 mM Tris pH 8, 150 mM NaCl, 1% Triton X-100) using a 2-mL potter-Elvehjem homogenizer and incubated on a rotating wheel for 30 min at 4 °C. Non-resuspended material was pelleted for 10 min at $20,000 \times g$ and 4 °C. The supernatant containing the fraction enriched in microsomal-associated proteins was transferred to a new tube and subjected to immunoprecipitation using the µMACS GFP isolation kit (Miltenyi Biotec). Immunoprecipitates were migrated on a 10% TGX stain-free polyacrylamide gel and SRF3-GFP protein levels detected using anti-GFP horseradish peroxidase-coupled antibodies (Miltenyi

Biotech 130-091-833, 1/5,000). Loading control was obtained using the stain-free that allows visualization of proteins in the gel. Signal intensity for SRF3-GFP was determined using Image J, relative to the stain-free loading control.

The protein extraction and western blot procedure followed the previous published method[65] with some modifications to perform MPK phosphorylation assay. Twenty seedlings of 6 days old light grown Col-0, srf3-3, and fls2 were treated in Fe sufficient liquid medium with 100 nM flg22 peptide for 5 or 10 min. The root material was cut and immediately frozen in liquid nitrogen and then was ground with liquid nitrogen and lysed directly in 100 µL 1x NuPAGE™ LDS Sample Buffer (Invitrogen™, Cat. NP0008) supplemented with 1x NuPAGE™ Sample Reducing Agent (Invitrogen™, Cat. NP0009) for 15 min on ice. The protein samples were denatured by heating for 10 min at 90 °C and centrifuge at $11,000 \times g$ for 10 min. The supernatant protein samples were separated by NUPAGE 10% Bis-Tris Plus Gel (Invitrogen™, Cat.NW00105BOX) and transferred onto Nitrocellulose membrane by iBlot 2 Dry Blotting system (Invitrogen™, Cat. IB23001). The phosphorylation status of MPK3,6 was detected by western blot with corresponding antibody (phospho-P44/42 MAPK antibodies (Cell Signaling, Cat. No. #4370) 1:2000 diluted in 1% BSA, Merck/Calbiochim, Cat. No.12657; Goat Anti-Rabbit IgG (H + L)-HRP Conjugate (Bio-Rad, Cat. No. 170-6515) 1:5000 in 5% non-fat milk). The same membrane was re-blotted with Tubulin antibody as the internal control (Invitrogen™, Cat. 32-2500, 1:5000 in 5% non-fat milk; Goat Anti-Mouse IgG (H + L)-HRP Conjugate (Bio-Rad, Cat. No. 170-6516) in 5% non-fat milk).

## Short-term iron deficiency, flg20, and flg22 treatments

Seeds were sowed in +Fe medium described in Gruber et al. and stratified for 2–3 days at 4 °C. 5 day old seedlings were treated for 4 h with low iron medium or for 2 h adding 100 µM of FerroZine or sufficiency in liquid medium described in Gruber et al. using 12-well plates[58]. Note that after the addition of ferrozine the pH was adjusted to the same pH = 5.7 as the control medium +Fe. However, no change in the pH was detected in agar adjusted with MES as described earlier and in Gruber et al.[58]. For flg22 treatment, Seeds were sowed in +Fe medium described in Gruber et al. and stratified for 2–3 days at 4 °C[58]. Five day old seedlings were treated for 4 h in iron sufficient media described in Gruber et al.[58] supplemented or not with flg22 or flg20.

## RhoNox-1 staining

Five day old seedlings were treated in ultra-pure distilled water (Fisher Scientific Invitrogen UltraPure Distilled Water 500 mL Plastic Container–10977015) called +Fe condition to get rid of any iron trace in water, 50 µM of ferrozine was added for 30 min. Then, the plants were transferred to ultra-pure distilled water supplemented with 2.5 µM of RhoNox-1 for 15 min (stock solution 5 mM; https://goryochemical.com/).

## Perls staining and DAB/H₂O₂ intensification

For Perls staining and DAB/$H_2O_2$ intensification, the embryos were dissected and isolated from dry seeds previously imbibed in distilled water for 3–4 h. The embryos were then vacuum infiltrated with Perls stain solution (equal volumes of 4% (v/v) HCl and 4% (w/v) K-ferrocyanide) for 15 min and incubated for 30 min at room temperature[66,67]. The DAB intensification was performed according to Meguro et al. as follows: after washing with distilled water, the embryos were incubated in a methanol solution containing 0.01 M $NaN_3$ and 0.3% (v/v) $H_2O_2$ for 1 h, and then washed with 0.1 M phosphate buffer (pH 7.4). For the intensification reaction the embryos were incubated between 10 to 30 min in a 0.1 M phosphate buffer (pH 7.4) solution containing 0.025% (w/v) DAB (Sigma), 0.005% (v/v) $H_2O_2$, and 0.005% (w/v) $CoCl_2$ (intensification solution). The reaction was terminated by rinsing with distilled water[68].

### GUS histochemical assay

Transgenic seedlings carrying *pCYP71A12:GUS* were grown on ½ MS media for 4 days and seedlings were then grown in 6-well plates containing ½ MS (+Fe or −Fe) liquid media for 16 h. Seedlings were then treated with 1 μM flg22 or 1 μM flg20 for 16 h. After treatment with peptides plants were washed with 50 mM sodium phosphate buffer, pH 7. One milliliter of GUS substrate solution (50 mM sodium phosphate, pH 7, 10 mM EDTA, 0.5 mM $K_4[Fe(CN)6]$, 0.5 mM $K_3[Fe(CN)6]$, 0.5 mM X-Gluc, and 0.01% Silwet L-77) was poured in each well. The plants were vacuum infiltrated for 5 min and then incubated at 37 °C for 4 h. Tissues were observed using a Discovery V8 microscope (Zeiss). Quantification of GUS signal in root tips of the stained seedlings was done using Fiji.

### Aniline blue staining

Five day old seedlings were incubated for 2 h in iron deficient or sufficient medium with or without DDG and then transferred for 2 h to 150 mM $K_2HPO_4$ and 0.01% aniline blue in 12-well plates wrapped in aluminum foil for light protection. Then imaging of the root epidermis in the elongation zone was performed.

### Sterol treatments

For inhibitor experiments, 5 day old seedlings were transferred to MS agar plates containing 50 μg/mL Fenpropimorph (https://www.caymanchem.com/; stock solution 50 μg/mL in DMSO) or 1 MM Lovastin (https://www.tocris.com/products/lovastatin_1530; stock solution 1 mM in DMSO) for 24 h.

### 2-Deoxy-D-glucose (DDG) treatment

Seedlings were grown on iron sufficient medium and after 5 days transferred to iron sufficient or low iron medium or flg22 containing medium with or without DDG (diluted in $H_2O$, stock 50 mM used at 50 μM; https://www.tocris.com/products/2-deoxy-d-glucose_4515).

For quantification methods and statistical analysis see Supplementary Methods.

### Reporting summary

Further information on research design is available in the Nature Research Reporting Summary linked to this article.

## Data availability

Source data are provided: https://github.com/mplatre/SRF3_Iron_Defense.git Raw reads used for the RNAseq are available at NCBI under accession number: PRJNA857745. The related count table is available on: https://www.scidb.cn/s/byui2e.

## Code availability

The code used in this paper are provided: https://github.com/mplatre/SRF3_Iron_Defense.git

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

## Acknowledgements

We thank Y. Belkhadir, B. Lacombe, I. Helariutta, and all the Busch lab members for critical discussions. Y. Jaillais for sharing cloning materials, F. Berger for providing H2B in PDONR P1P2, J.B.D. Long for providing SV40, Y. Benitez-Alfonso for providing 35s:PdBG1-mCITRINE line, I. Helariutta for providing feedback on the manuscript and sharing 35s:GFP-CALS3, cals3-3d, and cals3-6 as well as R. Stadler for providing 35s:DRONPA-s line. J. Y. Lee for providing 35s::PDLP5-GFP line. E. Bayer for providing pPDLP3::PDLP3-YFP and pSUC2::GFP lines. pCYP72A::GUS line was kindly provided by Y. Belkhadir. We thank, T. Zhang and the Salk Biophotonics core team for microscopy advance and assistance in quantification. We thank the Salk peptide synthesis core especially Jill Meisenhelder and finally Br. Moussu for thoughtful discussion. We would like to thank N. Gibbs and C.N. Miller for their careful reading of the manuscript. This study was funded by the National Institute of General Medical Sciences of the National Institutes of Health (grant number R01GM127759 to W. Busch), a grant from the Austrian Science Fund (FWF I2377-B25 to W. Busch), funds from the Austrian Academy of Sciences through the Gregor Mendel Institute (W. Busch), and start-up funds from the Salk Institute for Biological Studies (W. Busch). M.P. Platre was supported by a long-term postdoctoral fellowship (LT000340/2019 L) by the Human Frontier Science Program Organization. R.A. and J.E.P. were supported by The Max-Planck Society and Germany's Excellence Strategy CEPLAS (EXC-2048/1, Project 390686111). M.v.R. was funded by an IMPRS Ph.D fellowship. The European Research Council (ERC) under the European Union's Horizon 2020 research and innovation program (grant agreement No. 772103-BRIDGING) to E. Bayer with the EMBO Young Investigator Program to E. Bayer.

## Author contributions

M.P.P. was responsible of all experiments described in the manuscript except for: qRT-PCR from extreme accessions performed by M. Giovannetti, B.E. did dry seed embryo dissection, M.R., R.A., and J.P. generated and characterized the pSRF3-SRF3-GFP line, G.V. provided pIRT1-NLS-2xYPET line, J.N. performed western blot of SRF3 under low iron and flg22, S.B.S. was involved in phenotyping, GWAS data processing and analysis, and performing pCYP72A11::GUS experiment. C. Goeschl performed GWAS data plotting and GUS signal quantification, L.B. performed the selection and generation of transgenic lines, M. Gleason imaged SRF3 reporter lines, M.C. conducted qRT-PCR for immune genes under iron deficiency, C. Gaillochet and L.Z. performed RNAseq data analysis, M. Glavier performed SRF3 immunogold electron microscopy and M. Grison performed callose immuno-localization. M.P.P., S.B.S., and W.B. conceived the study and designed experiments. M.P.P., W.B., and E.B. wrote the manuscript, and all the authors discussed the results and commented on the manuscript.

## Competing interests

The authors declare no competing interests.
