## [Peer Review File · Nature Communications]

The receptor kinase SRF3 coordinates iron-level and flagellin dependent defense and growth responses in plantsREVIEWER COMMENTS

Reviewer #1 (Remarks to the Author):

In this study, Platre and colleagues raise evidence for a role of SRF3, a member of the leucine-rich repeats receptor kinase family, in iron (Fe) homeostasis and immunity in Arabidopsis roots. SRF3 was identified as the most likely causal gene for a significant association detected by GWAS for primary root growth under low external Fe. Compared to WT, primary root growth of *srf3* mutants is less significantly inhibited by -Fe. Analysis of reporter lines indicated that SRF3 promoter is mainly active in the elongation and differentiation zones, while the protein was surprisingly more abundant in the meristematic zone. The authors found that SRF3 is located at the plasma-membrane – in two sub-populations. Low Fe rapidly leads to SRF3 disappearance from the plasma-membrane, a process that can be prevented by removing the protein's extracellular domain or impairing the protein's kinase activity. With the help of callose immunostaining, aniline blue staining and a callose synthase inhibitor, the authors raised evidence that SRF3 modulates callose deposition in the root meristem. Although the SRF3-dependent regulation of callose synthases did not significantly affect cell-to-cell movement in different root zones, the authors found that it altered IRT1 expression and root growth regulation under low Fe. Interestingly, the authors demonstrate that, similar to low Fe, the elicitor peptide flg22 also inhibits SRF3 accumulation at the plasma-membrane and that *srf3* mutants are more resistant to flg22-induced root growth inhibition and IRT1 up-regulation. Finally, RNA-seq experiments indicated a surprisingly large overlap between early transcriptional responses to low Fe and to flg22, and that SRF3 coordinates a large part of the commonly-regulated genes.

The reported findings are novel and advance our understanding of how changes in external Fe levels are locally sensed in roots. The study also provides new mechanistic insights into how plants coordinate root responses to low Fe and to bacterial elicitors, which should be of interest to a broad audience. An impressive number of approaches was used to investigate different molecular aspects related to SRF3 function and to determine how the protein is controlled by Fe and flg22.

However, as detailed below, some of the conclusions drawn from the data are not so persuasive as the phenotypical differences are often minor and some findings are difficult to tie together. Furthermore, it remains difficult to reconcile how processes happening at different zones along the root are connected, especially when they occur within very short time spans.

Major comments:

1. The authors must be more careful when placing in the same model events that occur in spatially distinct parts of the root. For example, according to the authors (lines 187-189) SRF3 senses early changes in apoplastic signals associated with Fe depletion. However, Fe levels were only quantified in the differentiation zone, while SRF3 protein is mainly located in the meristem and this was the region always used to investigate Fe-dependent changes in SRF3 protein levels. Furthermore, the authors conclude that SRF3-dependent modulation of callose deposition tunes IRT1 expression but callose deposition was investigated only in the meristem, where IRT1 is not expressed. How many of these events do co-localize spatially and how many of them dependent on the function of quick signaling (by a yet unknown mechanism) connecting the meristem and the differentiation zone?

2. Although some aspects such as the existence of two SRF3 subpopulations at the plasma-membrane were investigated in great detail, the study leaves many important aspects unaddressed, as follows:
a) The link between natural variation and the rest of the story is weak: if SRF3 expression (at least at whole root level) is not significantly different among accessions and no non-synonymous substitution at the coding level is apparent, how to explain the association detected with GWAS? The authors should try to complement the *srf3* mutant with SRF3 expressed under promoters from different accessions. Furthermore, the authors should check if the deletion in SRF3's promoter affects the tissue- and root zone-specific expression of SRF3. Furthermore, how many of the "fast grower" accessions carry the deletion in SRF3's promoter?

b) The phenotypes of SRF3-OXs are not consistent: compared to WT, early root growth response to low Fe was increased (Fig. 2E), late root growth decreased (Suppl. Fig. 5E) while root growth for 3 days unchanged (Fig. 5D). Intriguingly, overexpression of the kinase dead mutant and the truncated

protein without the extracellular domain behave like WT both in the short and long term, even though these mutations prevent low-Fe-induced SRF3 disappearance from the plasma-membrane. How can these apparent inconsistencies be explained? Does SRF3 protein localization change when the gene is overexpressed or does it still mainly accumulate in the meristematic zone? Results for only one SRF3-OX line are shown whereas, at least the most critical experiments, at least three independent lines should be used.

c) The link between flg22-dependent changes in Fe and SRF3 is not sufficiently strong. The authors only show Fe localization in the differentiation zone. Since SRF3 promoter activity in this root zone is not affected by flg22 (Suppl. Fig. 8J), it is necessary to demonstrate if flg22 causes the same effect in the meristem, where SRF3 protein is present.

d) In order to better connect the SRF3 with the observed root growth phenotype, it is necessary to clearly determine if SRF3 affects also cell elongation besides affecting meristem size.

3. I would also like to see the quantitative results of root growth presented in a less processed manner. One example: although the authors explain in the Methods how they calculated "root growth response", which they expressed in arbitrary units, I cannot understand why the authors did not simply show mean root growth rate as mm day⁻¹? Even better would be to present time-course changes in mean growth rate (mm day⁻¹) from the day of transfer until the last recorded day. This would enable to estimate more directly the magnitude of the reported differences and when they became significant throughout the course of the experiment.

Minor comments:

1) To impose different levels of Fe deficiency, the authors supplied agar plates with increasing concentrations of the Fe(II) chelator Ferrozine. At such high levels, Ferrozine itself may have an inhibitory effect on root growth that could enhance the effect of Fe deficiency. To estimate a possible effect of Ferrozine, it would be important to validate at least some of the results, including the decreased SRF3 protein at the plasma-membrane, with the same concentration of Ferrozine (i.e. 100 μ M) but in the presence of Fe.

2) Furthermore, as different conditions were used to decrease external Fe levels in the media, it is necessary to indicate more clearly in the figures and/or figure legends what -Fe condition was used in each experiment. In many figures, only -Fe is indicated without mentioning if Ferrozine was or not present and at what concentration.

3) Is the strong zone-specific SRF3 protein localization affected by Fe availability and/or flg22?

4) The results of Fe staining with RhoNox-1 are intriguing. As this stain has not yet been commonly used in plants, the authors should at least validate the results of one experiment with Perls/DAB.

5) Is it possible to say if most RhoNox-1 signal was apoplastic? From the images it is hard to see.

6) Fig. 1D: how many Fe deficiency-induced genes were significantly differentially regulated in *srf3* compared to WT? Was the expression of IRT1 and FRO2 also altered in the mutant?

7) Is the SRF3 promoter activity eventually also decreased by Fe or flg22? If yes, would it be possible to track the time between transcription and protein accumulation in the two root zones by transferring plants back to non-stressed conditions?

8) Why were the cell-to-cell movement experiments with SUC2-GFP and DRONPA not carried out also after a more prolonged time? At 4h, the differences in callose deposition and aniline blue staining were not yet so prominent. However, perhaps impaired cell-to-cell movement can be detected at a later time-point.

9) In which cell-types of the meristem is SRF3 located? In all cell layers or mainly in the outer layers?

10) Lines 249-251: according to quantification of anti-callose fluorescence shown in Fig. 4A, callose was not significantly increased by -Fe.

11) To avoid confusion with SRF3-mCIT localization, the authors should use another color code to more clearly indicate that 3A shows pseudo-colored images.

12) Suppl. Fig. 3: SRF3 expression is shown for 3 slow grower accessions, but from these the root growth phenotype of only one is presented.

13) In Suppl Fig. 1A: "mm" is not a unit for "growth rate".

14) In Suppl Fig. 1H: Again, unclear unit. If what is shown is cm/day, then the data show growth rate and not simply growth. What exactly do the results show?

- 15) Lines 332-333: according to Tukey test, response of WT and *srf3-3* to *flg22* was not statistically different.
- 16) Fig. 1D: use "Fe sufficient condition" instead of "basal" in the y-axis.
- 17) Suppl. Fig. 8: in the legend, SEM is incorrectly described as "standard deviation of the mean". Please amend and indicate clearly whether the error bars indicate standard deviation or standard error.
- 18) Suppl. Fig. 8F-G: the variation of root lengths recorded for *bts-1* and *opt3-2* was surprisingly much higher than all other similar datasets. Were the mutants segregating? Or was there any other reason for the very large variation specifically in the experiments with these two mutants?
- 19) Lines 116-119: iron localization in seeds is not a parameter for seed Fe content or concentration. Therefore, rephrase to "levels of stainable Fe were not different between the mutant and WT".
- 20) In Suppl Fig. 4, please correct *col0* to *Col-0*. Use a consistent abbreviation for Columbia-0 (currently *Col0* and *Col-0* are used).

Reviewer #2 (Remarks to the Author):

The manuscript reports functional roles of an Arabidopsis receptor kinase SRF3 in iron homeostasis. Compared to wild-type *Col-0*, *srf3-3* mutant maintains higher iron content in roots when iron concentration in culture media is low. Consequently, *srf3-4* mutants induce less activity of a metal transceptor IRON-REGULATED TRANSPORT 1 (IRT1) promoter activity. The findings suggest that *srf3* mutants are more tolerant to iron deficiency than wild type at an early developmental stage. SRF3 is localized to the plasma membrane and plasmodesmata (PD), whereas low iron conditions induce the degradation of SRF3 protein. Interestingly, *srf3-3* mutant is more resistant to flagellin peptide *flg22* treatment compared to that of wild type. *flg22* also triggers degradation of SRF3 protein. In addition, around 90% of genes differentially regulation by low iron conditions represents around 7% of genes differentially regulated by *flg22*.

The authors stated that SRF3 is a negative regulator of callose deposition and coordinates iron-level and bacterial immunity. However, the data presented in the manuscript do not fully support the statement.

Functions of SRF3 in regulating callose synthase activity:

As show in Figure 4B, SRF3 doesn't seem to involve in callose accumulation at PD. The data shown in Figure 4C and 4D also support the assumption. Higher level of signals detected by anti-callose shown in Figure 4A might present higher callose accumulation in the root apoplast rather than PD.

If SRF3 is a negative regulator of callose synthases, one would expect that SRF3-OX would suppress short root phenotype in *cals3-3d* as stated. As the wild-type SRF3 might not be able to regulate *cals3-3d*, testing the genetic interaction between the genes using *srf3* and *cals3* (or higher order *cals* mutant) might be more straightforward. Also, it *cals* mutant might be a better option than using a callose synthase inhibitor, DDG.

From the provided data, it's not clear whether the PD localization of SRF3 required the extracellular domain or kinase activity. Both Figure 2D and 3A exhibit stronger punctate signals for SRF3DEXtraC and SRF3KD. It's suggested to image the fusion proteins with a PD marker or aniline blue-stained callose at PD.

If IRT1 regulation relies on early signaling events that are dependent on callose synthase, it's expected to observe higher expression level of IRT1 in *srf3* mutants. It seems that the expression level of SRF3 is regulated by iron concentration rather than callose accumulation. Does DDG treatment affect iron concentration in wild-type *Col-0*? One could assume that DDG treatment increases iron

accumulation in the apoplast, leading to the suppression of IRT1 expression.

Functions of SRF3 in flagellin dependent defense:

Compared to the roles of SRF3 in callose accumulation, the role of SRF3 in bacterial immunity seems more likely, However, there is no direct evidence to suggest the role of SRF3 in plant nutritional immunity in this manuscript. The findings on a similar SRF3 protein dynamic under low iron and flg22-treated conditions provide a promising link, but not sufficient to conclude that SRF3 is a point of convergence between iron and flg22-dependent signaling mediating root growth regulation.

Are srf3 mutants compromised in flg22-triggered immune responses?

Are srf3 mutants compromised in bacterial immunity?

Can SRF3 autophosphorylate and phosphorylate target proteins with or without bacterial stimulus?

What is the relationship between SRF3 and FLS2 (and other PRRs and co-receptors)?

Minor comments:

Line 98-103: The statements sounded like srf3 mutants are more sensitive to low iron levels, but the data suggest that srf3-3 is less sensitive to low iron in terms of both primary and secondary root growth. Please consider rephrasing the statements.

Are srf3 mutants contain higher intercellular Fe²⁺?

Line 139-140: Can srf3-3 mutant tolerant longer-term iron deficiency during different stages of plant development? As srf3 mutant was not included in the Figure 5SE, it's unable to determine whether SRF3-OX behaves differently to srf3 during the late response to low iron.

Line 184-186: The data only support the statement that the extracellular and kinase domains of SRF3 is required for its function. To demonstrate the fine regulation of SRF3 proteins accumulation at the PM, it is required to demonstrated that misregulation of functional SRF3 at the PM results in the root growth response to low ion conditions.

Figure 3B: The labeling of the figure suggests that PDLP3 and CALS3 are PM markers.

Line 237-238: Please rephrase the subtitle.

Line 241-245: Do Fen and Lova prevent the PD localization of SRF3? Also, the findings here can only suggest that SRF3 associated to PD through sterol not its functional role in PD.

Line 261: add (Figure S4B) following synthase inhibitor.

Line 327-329: The statement is unclear. Please consider rephrasing the sentence.

Line 307: SRF3-OXxcal3-3d > SRF3-OXxcals3-3d

Line 403-404: The fact that flg22 and low iron treatment doesn't change the SRF3 transcript level is not a supporting evidence for the role of SRF3 in signal transduction.

Line 430: It's unclear what a signaling function of callose synthases means?

Reviewer #3 (Remarks to the Author):

In this manuscript, the authors identified a receptor kinase gene SRF3 that was associated with the root growth under low iron levels through GWAS. They further provided experimental evidence for the role and mechanism of SRF3 in coordinating root growth, iron homeostasis and immunity pathways. Overall, this manuscript presents an interesting story.

However, several issues are also required for major revision.

1. The method of GWAS was used to detect the genes associated with root growth rate under low iron levels, and some SNPs were identified. Because the difference of phenotypic traits among genotypes is the base for GWAS, the authors should firstly illustrate that the population panel is suitable for GWAS analysis.
2. More information of GWAS analysis should be provided. The mixed linear model is used for GWAS analysis, according to the description in the section of methods. However, the population structure and kinship should also be provided. In addition, the result of association analysis is suggested to have high probability of false positive, please provide the methods in reducing false positive, for example, the p value threshold.
3. Association analysis is based on linkage disequilibrium (LD). The size of the mapping interval depends on the ratio of LD decay. All the genes in the LD decay interval should be taken into account. To confirm the candidate gene, all the other genes in the interval of LD decay should be excluded. In this manuscript, the T-DNA mutant was used to confirm the causal gene. However, the interval of LD decay and the genes in this interval should be provided. The detailed methods for excluding other genes should also be provided.
4. Results of significance tests should be provided, including the figures 3 and 6A. Please check the raw data and provide clear statistical results for all statistic data.
5. The manuscript should be further edited. Many sentences are too long, and it is very hard to get the important information.
6. It would be nice to provide a diagram for SRF3 to conclude its role and mechanism in the section of discussion.

Specific issues:

Introduction:

Lines 36-38: Where is the reference?

Lines 38-41: This sentence is too long, and hard to understand. Please change.

Lines 48-49: The reference should be provided.

Line 72: "is" -> "has been"

Results:

Lines 92-105:

I noticed that only one SNP with $-\log_{10}(\text{p-value})$ above 6 was shown in the black box of Manhattan plot in both the Figure 1A and S1B, but five SNPs in the magnified regions of Figure 1A. Please check the GWAS results, and confirm how many SNPs were identified to be associated with root growth rate under low iron levels. In addition, the percentage of phenotypic variations that total and candidate SNPs can explain should be provided, respectively.

Figure 1A showed four candidate genes in the association regions on chromosome 4. Why did you only investigate two genes SRF3 and DFL2? How did you exclude the other two candidate genes? $-\log(\text{p-value})$ in Figure 1A should be $-\log_{10}(\text{p-value})$.

Line 97: Please show the statistical results for the comparison of the relative expression levels between T-DNA mutant lines and WT in Figure S1D.

Lines 106-111: The statistical results in Figure 1D should be provided. To validate the expression profile from the RNA-seq data, qRT-PCR analysis is also required for examining the transcript levels of these DEGs.

Lines 113-116: I cannot follow the relationship between OPT3 and SRF3. RNAseq data also showed that the expression of OPT3 was up-regulated in *srf3* mutant?

Lines 153-154: "SRF3 transcript might expressed transiently in the meristematic cells." -> "SRF3 might express transiently in the meristematic cells".

Lines 160-164: This sentence is too long, please separate it into two sentences.

Line 257: The reference should be provided for CALS3.

Lines 275-278: Please check the ANOVA results. A decrease of signal in the activated cell and a concomitant increase in the surrounding cells were observed, but your statistical results in Figure 4D showed no differences among all comparisons.

Lines 292-297: The ANOVA results for Figure 5B should be provided.

Line 303: Please provide the reference for "PdBG1 known to negatively regulate callose deposition."

Line 307. Please clearly point out the SRF3-OXxcal3-3d line in the picture of Figure 5D.

Lines 327-329: What is the role of SRF3 in roots? Please clearly state the results from GO analysis of RNAseq data and GUS staining assays, respectively, and point out the role of SRF3 in roots here. In addition, the ANOVA analysis for GUS quantification should be performed.

Lines 332-333: Figure S8B showed that the difference of late root growth response to flg22 was not significant between WT and *srf3-3*, so how did you conclude that *srf3* roots were only impaired in their response to flg22? Please re-analyze the raw data or revise this result.

Line 337: "higher iron root content" -> "higher iron content in the root"

Lines 352-353: The ANOVA results for Figure 6E should be provided.

Discussion:

The authors spent much time on concluding and repeating the results. It would be nice to point out and discuss several important novel insights of this manuscript. English writing has to be improved. More importantly, can you provide a diagram for SRF3 to explain its role and mechanism in coordinating root growth, iron homeostasis and bacterial immune responses?

Methods

Lines 717-730: Please provide more detailed methods for GWAS analysis.

Line 794: I could not find the reference for Platre et al, 2019.

Lines 832, 888, 895 and 976: Please provide the reference for Gruber et al., 2013.

Lines 904 and 907: The reference for Roschztardt et al., 2009 and Stacey et al., 2008, should be provided, respectively.

Lines 1133 and 1142: Please provide the reference for Vert et al., 2002 and Nakagawa et al, 2007, respectively.

REVIEWER

COMMENTS

We thank the reviewers for appreciating the relevance and interest of our work, and for providing us with constructive criticisms and suggestions. We have now substantially revised our manuscript and responded to all points raised by the reviewers. We did this by conducting several challenging experiments that have taken several months to complete, and by clarifying several issues in the text. We think that based on the reviewers' feedback our manuscript has greatly improved and are grateful for that.

Reviewer #1 (Remarks to the Author):

In this study, Platre and colleagues raise evidence for a role of SRF3, a member of the leucine-rich repeats receptor kinase family, in iron (Fe) homeostasis and immunity in Arabidopsis roots. SRF3 was identified as the most likely causal gene for a significant association detected by GWAS for primary root growth under low external Fe. Compared to WT, primary root growth of *srf3* mutants is less significantly inhibited by -Fe. Analysis of reporter lines indicated that SRF3 promoter is mainly active in the elongation and differentiation zones, while the protein was surprisingly more abundant in the meristematic zone. The authors found that SRF3 is located at the plasma-membrane – in two sub-populations. Low Fe rapidly leads to SRF3 disappearance from the plasma-membrane, a process that can be prevented by removing the protein's extracellular domain or impairing the protein's kinase activity. With the help of callose immunostaining, aniline blue staining and a callose synthase inhibitor, the authors raised evidence that SRF3 modulates callose deposition in the root meristem. Although the SRF3-dependent regulation of callose synthases did not significantly affect cell-to-cell movement in different root zones, the authors found that it altered IRT1 expression and root growth regulation under low Fe. Interestingly, the authors demonstrate that, similar to low Fe, the elicitor peptide flg22 also inhibits SRF3 accumulation at the plasma-membrane and that *srf3* mutants are more resistant to flg22-induced root growth inhibition and IRT1 up-regulation. Finally, RNA-seq experiments indicated a surprisingly large overlap between early transcriptional responses to low Fe and to flg22, and that SRF3 coordinates a large part of the commonly-regulated genes.

The reported findings are novel and advance our understanding of how changes in external Fe levels are locally sensed in roots. The study also provides new mechanistic insights into how plants coordinate root responses to low Fe and to bacterial elicitors, which should be of interest to a broad audience. An impressive number of approaches was used to investigate different molecular aspects related to SRF3 function and to determine how the protein is controlled by Fe and flg22.

However, as detailed below, some of the conclusions drawn from the data are not so persuasive as the phenotypical differences are often minor and some findings are difficult to tie together. Furthermore, it remains difficult to reconcile how processes happening at different zones along the root are connected, especially when they occur within very short time spans.

Major comments:

1. The authors must be more careful when placing in the same model events that occur in spatially distinct parts of the root. For example, according to the authors (lines 187-189) SRF3 senses early changes in apoplastic signals associated with Fe depletion. However, Fe levels were only quantified in the differentiation zone while SRF3 protein is mainly located in the meristem and this was the region always used to investigate Fe-dependent changes in SRF3 protein levels. Furthermore, the authors conclude that SRF3-dependent modulation of callose deposition tunes IRT1 expression but callose deposition was investigated only in the meristem, where IRT1 is not expressed. How many of these events do co-localize spatially and how many of them dependent on the function of quick signaling (by a yet unknown mechanism) connecting the meristem and the differentiation zone?

We thank the reviewer for pointing out these important issues. To address this, we have now calculated the iron levels in the elongation zone which corresponds to the region where SRF3 is degraded upon low iron levels (Figure 1f, 2c and Supplementary Figure 2a-c and Movie 5). Moreover, this zone corresponds to the region where we observed defects for *srf3-3* in decreasing cell elongation-mediated root growth under low iron levels (Supplementary Figure 4r). We also studied callose deposition in the meristem (Figure 4A) and the elongation zone with Aniline blue staining (Figure 4b). Moreover, IRT1 is expressed in the elongation zone until the transition zone after 24 hours of exposure to low iron as shown in Figure 1e. In order to establish whether this mechanism might not be restricted to the root tip, we have now shown that low iron triggered SRF3 degradation is not limited to the root elongation zone by conducting western blot on total root extract (Supplementary Figure 5c). Taken together, we have now shown that in the elongation zone early lack of iron concomitantly a) decreases cellular iron levels, b) triggers SRF3 degradation, c) mediates SRF3-dependent root cell length decrease, d) triggers callose signaling, e) leads later-on to IRT1 activation by an unknown mechanism.

2. Although some aspects such as the existence of two SRF3 subpopulations at the plasma-membrane were investigated in great detail, the study leaves many important aspects unaddressed, as follows:

a) The link between natural variation and the rest of the story is weak: if SRF3 expression (at least at whole root level) is not significantly different among accessions and no non-synonymous substitution at the coding level is apparent, how to explain the association detected with GWAS? The authors should try to complement the *srf3* mutant with SRF3 expressed under promoters from different accessions. Furthermore, the authors should check if the deletion in SRF3's promoter affects the tissue- and root zone-specific expression of SRF3. Furthermore, how many of the "fast grower" accessions carry the deletion in SRF3's promoter?

We have used natural variation as a screening tool for new regulators of the growth response to low iron levels. This resulted in the discovery of SRF3 as a novel regulator of growth responses to low iron levels. Because, to our knowledge no other LRR-RK had been directly implicated in the response to low iron levels, we focused on characterizing the signaling and molecular roles of SRF3 in this response. This resulted in a set of a large number of comprehensive experiments that elucidated a novel molecular mechanism for the control of growth, iron homeostasis and defense pathways. Our focus and the significant time and resources that we had to leverage for this, only allowed for a limited test of whether SRF3 alleles are truly causal for the observed natural variation. We aimed to test whether *SRF3* allelic variation in controlling root growth under low iron using a complementation approach. Unfortunately, this did not include the deletion (when we started the cloning, we didn't have the deletion on the radar). We did not observe a clear correlation between the sequences of the fast or slow growers and the root growth rate under low iron levels (Supplementary Fig. 3e,f). Therefore, while tempting to speculate that SRF3 variants and in particular the deletion might be involved in determining root growth rate in low iron conditions in accessions, its roles need further investigation. Due to the pandemic, our limited funding and the plethora of interesting experiments to unravel SRF3's molecular function for its role in the growth regulation, we could not pursue additional experiments and have to leave this up to further studies. We have included these issues in the manuscript text and hope that the reviewers agree that the role of SRF3, the novel regulatory mechanism and its complexity are more than enough for one manuscript.

b) The phenotypes of SRF3-OXs are not consistent: compared to WT, early root growth response to low Fe was increased (Fig. 2E), late root growth decreased (Suppl. Fig. 5E) while root growth for 3 days unchanged (Fig. 5D). Intriguingly, overexpression of the kinase dead mutant and the truncated protein without the extracellular domain behave like WT both in the short and long term, even though these mutations prevent low-Fe-induced SRF3 disappearance from the plasma-membrane. How can these apparent inconsistencies be explained? Does SRF3 protein localization change when the gene is overexpressed, or does it still mainly accumulate in the meristematic zone? Results for only one SRF3-OX line are shown whereas, at least the most critical experiments, at least three independent lines should be used.

These are very important points and we have tried to clarify this better in the text. Fig2e and Supplementary Figure 5f correspond to the response of SRF3-OX to the lack of iron. However, in Figure 5d the observed values correspond to root growth rate under iron sufficient media (+Fe). We think that the observed responses are consistent with the early and late root growth responses to low iron being distinct and a role of SRF3 as a scaffold protein (triggering a change of its stoichiometry overexpressing it affects the protein function and related phenotypes either positively or negatively). We have clarified this in the text (line 201-203). Concerning the SRF3-OX kinase dead and the version lacking the extracellular domain, the root growth phenotypes are similar to WT for short and long-term responses to low iron. Similarly, to the OX construct, these constructs are in the WT background. The WT-like response therefore indicates that when the functions of these important domains are disrupted, they don't interfere with the function of the WT version of the protein and the

stoichiometry of any complex. For more clarity, this section has been developed in the text (line 208-210).

In order to further test and corroborate the effect of these overexpressing lines, we have conducted numerous additional experiments. Overexpression of SRF3 seems not to affect SRF3 protein localization and a new figure with this has been added (Supplementary Figure 4g). We also have added RT-PCR of two independent transgenic lines for each construct and performed long-term transfer assays to low iron medium with these lines (Supplementary Figure 5f-g). Furthermore, we have now performed three additional independent early root growth assays that include one insertion line of SRF3-OX^{WT} and the mutated versions. We obtained the same results as previously: SRF3-OX^{WT} remained insensitive compared to WT and the SRF3-OX of mutated versions under low iron levels. We also have now included the late root growth response of two independent experiments with two independent insertion lines and observed the same trend as previously. However, for the early and late root growth of SRF3-OX^{KD} for the line #25 and #14, respectively, we observed that the phenotype is partially rescued to the WT level suggesting that SRF3 kinase activity might be required for this response to a given extent (Supplementary Figure 5f-g). This prompted us to test whether SRF3 has an *in vitro* kinase activity. We performed protein extraction from plants expressing SRF3^{WT}, SRF3^{KD} and BRI1 as a positive control. From there, we could not conclude if SRF3 possesses a kinase activity (figure below). Since this experiment is inconclusive, we did not add it to the main manuscript, but we rephrased carefully our conclusions talking about the kinase domain rather than kinase activity and further discussed its plausible kinase activity (line 212-214 and 226-227).

Legend: Left blot, input protein extracted after immunoprecipitation with GFP-trap using anti-GFP antibody. Red dash boxes indicate the size of BRI1 and SRF3^{WT} and SRF3^{KD}. Right, blotting using anti-ATP after performing in vitro kinase assay. Red dash boxes indicate the size of BRI1 and SRF3^{WT} and SRF3^{KD} showing their auto phosphorylated form. However, this is not possible to conclude for SRF3 since the band seems to be not specific as the band is observed in BRI1, green dash box.

c) The link between flg22-dependent changes in Fe and SRF3 is not sufficiently strong. The authors only show Fe localization in the differentiation zone. Since SRF3 promoter activity in this root zone is not affected by flg22 (Suppl. Fig. 8J), it is necessary to demonstrate if flg22 causes the same effect in the meristem, where SRF3 protein is present.

Using data obtained via confocal microscopy, we now have calculated in two different backgrounds SRF3 protein levels in the elongation zone (Figure 6c and Supplementary Figure 9i) and perform western blots for SRF3 on total root extract upon flg22 (Supplementary Figure 5c). Moreover, iron levels in the elongation zone decrease upon application of flg22 for 1 hour. These data unambiguously show that SRF3 degradation occurs within 2-4 hours upon flg22 treatment in the elongation zone.

d) In order to better connect the SRF3 with the observed root growth phenotype, it is necessary to clearly determine if SRF3 affects also cell elongation besides affecting meristem size.

We have measured the cumulative cell length from the meristem to the transition zone (26th cell) and found that under low iron medium root cells are shorter since the curve is below the one under iron sufficient condition in the WT but not in *srf3* mutants (Figure S4E-F). Moreover, the steepness of the curve was identical in both conditions in WT meaning that the meristem is not smaller under low iron conditions (Figure S4E-F). To further assess the meristem size early under low iron levels, we have now calculated the size of the meristem based on the distance between the quiescent center to the cell for which the length is longer than the width in the cortex cell files and observed no difference (Supplementary Figure 4q). This result suggested that under low iron levels, root growth is decreased through a change in cell elongation in a SRF3-dependent manner. This is further supported by the measurement of the fourth elongated cells, which was shorter in WT in low iron levels, but no difference was observed in *srf3* mutants (Supplementary figure 4r). All these experiments indicate that the cell division is not significantly affected during the early response to low iron levels and that the decrease of the root growth is mainly due to an SRF3-dependent reduction of cell elongation. More clarity has been provided in the text legend to explain this result (Line 174-177 and legend Supplementary figure 4).

3. I would also like to see the quantitative results of root growth presented in a less processed manner. One example: although the authors explain in the Methods how they calculated "root growth response", which they expressed in arbitrary units, I cannot understand why the authors did not simply show mean root growth rate as mm day⁻¹? Even better would be to present time-course changes in mean growth rate (mm day⁻¹) from the day of transfer until the last recorded day. This would enable to estimate more directly the magnitude of the reported differences and when they became significant throughout the course of the experiment.

We have provided in the Figure S1, the raw data as well as the time-course over three days. However, we choose to present the data using the root growth response in the rest of the paper since in our opinion, the growth responses to low iron are easier to compare when presented as response as the figures are less packed.

Minor comments:

1) To impose different levels of Fe deficiency, the authors supplied agar plates with increasing concentrations of the Fe(II) chelator Ferrozine. At such high levels, Ferrozine itself may have an inhibitory effect on root growth that could enhance the effect of Fe deficiency. To estimate

a possible effect of Ferrozine, it would be important to validate at least some of the results, including the decreased SRF3 protein at the plasma-membrane, with the same concentration of Ferrozine (i.e. 100 μ M) but in the presence of Fe.

We thank the reviewer for this suggestion. We studied the decrease of SRF3 at the plasma membrane as well as the early root growth response without Ferrozine. For both time lapse analyses of SRF3 protein levels (Figure 2C) and root growth (Figure 2A) the ferrozine has been removed confirming that the effect is not specific of Ferrozine. We have now edited the text to avoid any confusion and indicated that when "-Fe" is written it means that no ferrozine was added to the low iron medium while when we write -FeFZ[X], X concentration ferrozine was present in low iron medium.

2) Furthermore, as different conditions were used to decrease external Fe levels in the media, it is necessary to indicate more clearly in the figures and/or figure legends what -Fe condition was used in each experiment. In many figures, only -Fe is indicated without mentioning if Ferrozine was or not present and at what concentration.

We have now edited the text to avoid any confusion and indicated that when "-Fe" is written it means that no ferrozine was added to the low iron medium while when we write -FeFZ[X], X concentration ferrozine was present in low iron medium (line 186).

3) Is the strong zone-specific SRF3 protein localization affected by Fe availability and/or flg22?

As our focus was directed on explaining the lack of root growth response in the *srf3* mutant under flg22 and -Fe which was due to cell elongation, we focused on SRF3 protein dynamic in the elongation zone. Currently, we can't exclude that the two treatments induce change of SRF3 protein localization in other cell types or in different zones in the root as these conditions trigger SRF3 degradation in the entire root as detected by western blot (Supplementary Figure 5c).

4) The results of Fe staining with RhoNox-1 are intriguing. As this stain has not yet been commonly used in plants, the authors should at least validate the results of one experiment with Perls/DAB.

We now have performed Perls/DAB staining in *srf3-3* mutant along with two positive controls, *bts-1* and *opt3-2* and observed the expected accumulation of iron in roots, such as meristematic, elongation and differentiation zones (Supplementary Figure 2d). This cross-validates the results from the RhoNox-1 staining method.

5) Is it possible to say if most RhoNox-1 signal was apoplastic? From the images it is hard to see.

In our manuscript, we provide evidence that SRF3 senses extracellular signal(s) which are associated with a fast decrease of iron levels. However, we can't establish with certainty if the signal is emitted by a decrease of Fe^{2+} or Fe^{3+} pools that might be intracellular or extracellular. RhoNox-1 is a fluorescent probe which detects the labile Fe^{2+} which is involved in the Fenton reaction to decrease ROS ($Fe^{2+} + H_2O_2 \rightarrow Fe^{3+} + OH \cdot + OH^-$). This probe has been characterized to localized in the cytoplasm, golgi and ER in rat retinal carcinoma^{1,2}. In line with this, we don't think that most of the RhoNox-1 reported signal is apoplastic as RhoNox-1 does not accumulate in the apoplast in the elongation zone when doing colocalization with a PM marker (See below). We observed in the merged picture only purple color at the PM instead of white which should raise colocalization between purple and green channels. We used PM marker, because at this confocal resolution we could not differentiate between the PM and apoplast markers.

Regardless of the source of the measured Rho-Nox1 signal, we think that Fe^{2+} is indeed present in plant apoplast since according to the literature H_2O_2 , the Fenton reaction and iron reduction by FRO2 occur in the apoplast/extracellular matrix³⁻⁵. Moreover, apoplastic iron can be detected by ICP-MS⁶.

Left: RhoNox-1. Middle: PM marker. Right: Merge. Colocalization can be seen in white

6) Fig. 1D: how many Fe deficiency-induced genes were significantly differentially regulated in *srf3* compared to WT? Was the expression of IRT1 and FRO2 also altered in the mutant?

The genes shown in Figure 1s and Supplementary Table 2 are the genes that are misregulated and that are known to be involved in the canonical iron deficiency. We noticed that other genes involved in the regulation of FIT are also mis regulated in *srf3* mutants (e.g. MYC2, ACS6 and EBF1, which are linked to jasmonic and ethylene pathway⁷⁻⁹). We only found IRT1 to be significantly up-regulated in iron sufficient condition in *srf3-3* but not in *srf3-2*. This is why it is not contained in Fig 1d. For FRO2, we did not observe any difference at the transcript level, but consistent with the upregulation of IRT1 upregulation and the ectopic iron accumulation in roots, FRO2 activity is higher in *srf3-3* mutant as measured by ferric reductase activity on plates. See quantification below (right).

Ferric chelate assay. From left to right: WT, *srf3-3* and *srf3-2*

7) Is the SRF3 promoter activity eventually also decreased by Fe or flg22? If yes, would it be possible to track the time between transcription and protein accumulation in the two root zones by transferring plants back to non-stressed conditions?

Based on the signal intensity level in the *SRF3* transcriptional reporter line (*pSRF3::mCITRINE-NLS-mCITRINE*) in the elongation zone, we have not observed a difference between +Fe and -FeZ[100] for 2 hours and between flg20[1] and flg22[1] for 4 hours, as well as in the control line (*pUBQ10::H2B-mSCARLET*; Supplementary Figure 5a and Supplementary Figure 9j). However, when we measured the distance between the quiescent center to the first cell showing a high signal intensity in the nucleus we observed that after each treatment, -FeZ[100] and flg22[1] compared to the control conditions this distance was reduced. Altogether, this suggests that both -FeZ[100] and flg22[1] might slightly activate the transcription of *SRF3* in the elongation-transition zone. This might indicate that upon -Fe and flg22 in the elongation zone, *SRF3* protein levels decrease while the transcription is increased. Thus, *SRF3* protein levels decrease is due to post transcriptional regulation independent of the transcription level.

8) Why were the cell-to-cell movement experiments with SUC2-GFP and DRONPA not carried out also after a more prolonged time? At 4h, the differences in callose deposition and aniline blue staining were not yet so prominent. However, perhaps impaired cell-to-cell movement can be detected at a later time-point.

In this paper, we investigated the role of *SRF3* role in the immediate growth response to a change in iron levels. This is because we have found that -Fe triggers a significant decrease of *SRF3* protein levels within an hour (Figure 2C) and within 4 hours for flg22 (Figure 6C). Fe and flg22 trigger a significant root growth decrease later-on after 4 hours and 8 hours,

respectively (Figure 2A and Figure 6B). Observations of IRT1 promotor activity revealed that -Fe and flg22 trigger its activation within 4 hours (Figure 5a and Supplementary Figure 10a). Thus, we focused our observations on this time-period because the signaling events mediating root growth arrest and IRT1 promotor activation need to occur in the time between the degradation of the SRF3 protein and the observation of a significant root growth decrease and *pIRT1* activation.

Regardless, we extended our observation in response to this reviewer's question and measured the normalized mean intensity in the epidermis and cortex for pSUC2 in a time lapse series from 2 hours to 12 hours after transfer to -Fe. We did not detect any significant decrease of the signal after 12 hours, suggesting that in 12 hours the -Fe-triggered callose deposition is not able to modify cell-to-cell communication of macro molecules.

9) In which cell-types of the meristem is SRF3 located? In all cell layers or mainly in the outer layers?

According to scRNAseq data, SRF3 expression can be detected in all cell layers. The data are provided in Supplementary Figure 4j-n.

10) Lines 249-251: according to quantification of anti-callose fluorescence shown in Fig. 4A, callose was not significantly increased by -Fe.

We have now performed a two ways Student test which indicates significance. The new test results have been added to the main figure (Figure 4a).

11) To avoid confusion with SRF3-mCIT localization, the authors should use another color code to more clearly indicate that 3A shows pseudo-colored images.

Thank for the comment. The change has been done accordingly to the request and the scale of the color-coded image added along with arrows indicating the plasma membrane associated dots (Figure 3a).

12) Suppl. Fig. 3: SRF3 expression is shown for 3 slow grower accessions, but from these the root growth phenotype of only one is presented.

We now present the expression data from 3 slow growing and three fast growing accessions in Supplementary Figure 3e, as well as complementation for all these accessions in Supplementary Figure 3f. We note that the result from these experiments (as stated earlier) is somewhat inconclusive regarding the causality of SRF3 for the natural variation.

13) In Suppl Fig. 1A: "mm" is not a unit for "growth rate".

It has been modified accordingly. Thanks.

14) In Suppl Fig. 1H: Again, unclear unit. If what is shown is cm/day, then the data show growth rate and not simply growth. What exactly do the results show?

It has been modified accordingly. Thanks.

15) Lines 332-333: according to Tukey test, response of WT and srf3-3 to flg22 was not statistically different.

We would like to thank the review for this comment since it was a mistake while annotating the figure. It has been corrected and added to Supplementary Figure 9b. Please find the result of the test below.

Means charts:

Genotype / Tukey (HSD) / Analysis of the differences between the categories with a confidence interval of 95% (Response):

Contrast	Difference	Standardized difference	Critical value	Pr > Diff	Significant
srf3-3_RESP vs 14-19_RESP	0.101	4.190	2.370	0.000	Yes
srf3-3_RESP vs WT_RESP	0.069	3.283	2.370	0.011	Yes
WT_RESP vs 14-19_RESP	0.033	1.256	2.370	0.353	No
Tukey's d critical value:			3.352		

Category	LS means	Standard error	Lower bound (95%)	Upper bound (95%)	Groups
srf3-3_RESP	0.958	0.017	0.925	0.991	A
WT_RESP	0.889	0.016	0.857	0.922	B
14-19_RESP	0.857	0.017	0.823	0.890	B

14-19 corresponds to srf3-3 comp

16) Fig. 1D: use "Fe sufficient condition" instead of "basal" in the y-axis.

It has been modified accordingly. Thanks.

17) Suppl. Fig. 8: in the legend, SEM is incorrectly described as "standard deviation of the mean". Please amend and indicate clearly whether the error bars indicate standard deviation or standard error.

It has been modified accordingly. Thanks.

18) Suppl. Fig. 8F-G: the variation of root lengths recorded for bts-1 and opt3-2 was surprisingly much higher than all other similar datasets. Were the mutants segregating? Or

was there any other reason for the very large variation specifically in the experiments with these two mutants?

The mutants have been sequenced and are not segregating. We don't know why the variation is high for these two mutants.

19) Lines 116-119: iron localization in seeds is not a parameter for seed Fe content or concentration. Therefore, rephrase to "levels of stainable Fe were not different between the mutant and WT".

It has been modified accordingly. Thanks.

20) In Suppl Fig. 4, please correct col0 to Col-0. Use a consistent abbreviation for Columbia-0 (currently Col0 and Col-0 are used).

It has been modified accordingly. Thanks.

Reviewer #2 (Remarks to the Author):

The manuscript reports functional roles of an Arabidopsis receptor kinase SRF3 in iron homeostasis. Compared to wild-type Col-0, *srf3-3* mutant maintains higher iron content in roots when iron concentration in culture media is low. Consequently, *srf3-4* mutants induce less activity of a metal transceptor IRON-REGULATED TRANSPORT 1 (IRT1) promoter activity. The findings suggest that *srf3* mutants are more tolerant to iron deficiency than wild type at an early developmental stage. SRF3 is localized to the plasma membrane and plasmodesmata (PD), whereas low iron conditions induce the degradation of SRF3 protein. Interestingly, *srf3-3* mutant is more resistant to flagellin peptide flg22 treatment compared to that of wild type. flg22 also triggers degradation of SRF3 protein. In addition, around 90% of genes differentially regulation by low iron conditions represents around 7% of genes differentially regulated by flg22.

The authors stated that SRF3 is a negative regulator of callose deposition and coordinates iron-level and bacterial immunity. However, the data presented in the manuscript do not fully support the statement.

Functions of SRF3 in regulating callose synthase activity:

As show in Figure 4B, SRF3 doesn't seem to involve in callose accumulation at PD. The data shown in Figure 4C and 4D also support the assumption. Higher level of signals detected by anti-callose shown in Figure 4A might present higher callose accumulation in the root apoplast rather than PD.

We thank the reviewer for pointing this out. We now have evaluated callose location using immunogold electron microscopy using anti-callose antibody upon low iron levels (-Fe without ferrozine) for 4 hours and have included the data in Figure 4c. In order to make sure that during this treatment SRF3 is removed from the PD and bulk PM, we have used the line *pSRF3::SRF3-GFP* (Ler) for which the decrease of SRF3 in the two fractions has been recorded (Figure 3C). We observed a slight but insignificant increase of gold particles at the PD (p-value 0.06), while no detectable increase in the bulk PM was noticed. Overall, these results weren't able to narrow down where the callose will be deposited and we discuss this in the text (line 282-285).

If SRF3 is a negative regulator of callose synthases, one would expect that SRF3-OX would suppress short root phenotype in *cals3-3d* as stated. As the wild-type SRF3 might not be able to regulate *cals3-3d*, testing the genetic interaction between the genes using *srf3* and *cals3* (or higher order *cals* mutant) might be more straightforward. Also, it *cals* mutant might be a better option than using a callose synthase inhibitor, DDG.

We thanks to the reviewer for raising this point, which we agree upon. We now have emphasized in the text that our assumption is that SRF3 regulate CALS signaling rather than its catalytic activity. To address this, we performed an experiment in which we overexpressed *SRF3* in the *CALS3-OX* and *cals3-6* mutant lines which had already been characterized¹⁰. *CALS3-OX* by itself is hyposensitive to low iron levels while *cals3-6* is hypersensitive, confirming that *CALS3* is required for regulating root growth under low iron levels (Figure 5e,f). The cross between *cals3-6* and *SRF3-OX* resulted in a hyposensitive root growth response while both single lines were hypersensitive, showing a genetic interaction of *CALS* and *SRF3*. Finally, overexpressing *SRF3* in *CALS3-OX* yielded a hyposensitive response to low iron, similar to *CALS3-OX* and opposite to *SRF3-OX* when compared to WT. This shows that *SRF3* is acting upstream of *CALS3* (Figure 5e,f and line 334-343).

From the provided data, it's not clear whether the PD localization of SRF3 required the extracellular domain or kinase activity. Both Figure 2D and 3A exhibit stronger punctate signals for SRF3DExtraC and SRF3KD. It's suggested to image the fusion proteins with a PD marker or aniline blue-stained callose at PD.

We thank the reviewer for pointing out that we presented this point not sufficiently clear. We have now changed the figure 3a, false colored it and added arrows to visualize this difference more clearly. We performed aniline blue staining, which further showed that SRF3DExtraC is not associated with punctate signals at the PM (Supplementary Figure 6d). Moreover, calculating the standard deviation of the mean intensity (SDMI) at the apical/basal part of the PM showed a decrease of this value confirming the loss of punctuated structure in this region of the PM (Figure 3a and Supplementary Figure 6b). However, we agree that SRF3^{KD} is still associated with the punctuated structures but to a lesser extent than the WT since the SDMI revealed a decrease of this value in Figure 3a and Supplementary Figure 6b. We therefore did not conclude strongly in the manuscript that the SRF3 kinase domain is required for its association with the punctuated structure but might be involved in the regulation of its partitioning between the punctuated structure and the bulk PM (line 230-232).

If IRT1 regulation relies on early signaling events that are dependent on callose synthase, it's expected to observe higher expression level of IRT1 in *srf3* mutants. It seems that the expression level of SRF3 is regulated by iron concentration rather than callose accumulation. Does DDG treatment affect iron concentration in wild-type Col-0? One could assume that DDG treatment increases iron accumulation in the apoplast, leading to the suppression of IRT1 expression.

Thanks for raising this good point. Performing treatment with DDG for 4 hours at 50uM, we observed no statistical differences with the mock condition in WT confirming that DDG does not impact root iron levels and thus *IRT1* expression levels (Supplementary Figure 8a).

Functions of SRF3 in flagellin dependent defense:

Compared to the roles of SRF3 in callose accumulation, the role of SRF3 in bacterial immunity seems more likely. However, there is no direct evidence to suggest the role of SRF3 in plant nutritional immunity in this manuscript. The findings on a similar SRF3 protein dynamic under low iron and flg22-treated conditions provide a promising link, but not sufficient to conclude that SRF3 is a point of convergence between iron and flg22-dependent signaling mediating root growth regulation.

Are *srf3* mutants compromised in flg22-triggered immune responses?

Yes, *srf3* mutants are impaired in flg22-triggered immunity. We provided evidence of this through the assay based on GUS staining reporting pCYP71A12 promoter activity (Figure 6a). In *srf3-4* the GUS signal was restricted to the root tip while in the WT it was spread out to the elongation-differentiation zone arguing for a role of SRF3 in flg22-mediated immune response (Figure 6a). Moreover, we performed an RNAseq experiment after two hours and observed that in the WT about 3600 genes were differentially expressed while in *srf3-2* and *srf3-3* only 250 genes were differentially expressed upon flg22 treatment for 2 hours, showing that *SRF3* controls the transcriptional landscape under flagellin perception (Figure 6f). From these two independent experiments we can conclude that *SRF3* is critical in regulating the response to flg22 and is acting as a positive regulator of this response. To provide more details about the mode of action of SRF3 on flg22-dependent immune response we have now performed a phosphorylation assay of MPK3/MPK6 which has been added to Supplementary Figure 10i. From this experiment, we could conclude that SRF3 acts independently of MPK3/MPK6 and might be involved in controlling BKI1/RBOH-triggered immunity.

Are *srf3* mutants compromised in bacterial immunity?

We found that *SRF3* is required for flagellin-elicited immune responses. We think that answering how this relates to bacterial immunity will be really interesting but further investigation was in our opinion out of the scope of this already very extensive and complex manuscript as it already covers a very extensive scope (identifying SRF3 as the regulator of a new, fast growth response to low iron levels, iron homeostasis and immune responses, uncovering the cause of this growth response to be removal of SRF3 from the membrane, finding the interaction with callose synthase as an important part of this new regulatory mechanism, elucidating the relation of low iron and flagellin elicited responses, etc.).

Can SRF3 autophosphorylate and phosphorylate target proteins with or without bacterial stimulus?

This is a very interesting question and would provide more details about SRF3 mode of actions. To this end, we now have provided more details about the mode of action of SRF3 on flg22-dependent immune response performing a phosphorylation assay of MPK3/MPK6 which has been added in Supplementary Figure 10i. From this experiment, we could conclude that SRF3 acts independently of MPK3/MPK6 and thus might not be involved in FLS2-mediated MPK3/MPK6 phosphorelay. A more comprehensive investigation of phosphorylation events was in our opinion out of the scope of this already very extensive and complex manuscript as it already covers a very extensive scope (identifying SRF3 as the regulator of a new, fast growth response to low iron levels, iron homeostasis and immune responses, uncovering the cause of this growth response to be removal of SRF3 from the membrane, finding the interaction with callose synthase as an important part of this new regulatory mechanism, elucidating the relation of low iron and flagellin elicited responses, etc.). Since we didn't experimentally address the kinase activity of SRF3, we carefully rephrased our conclusions and refer to the kinase domain rather than the kinase activity and further discussed its plausible kinase activity (line 212-214).

What is the relationship between SRF3 and FLS2 (and other PRRs and co-receptors)?

As discussed in reply to the previous comment, we don't think a comprehensive study regarding the interaction of FLS2 and other PRRs and co-receptors is in the scope of this paper. Since this will be an important line of further investigation, we included this point in the discussion (line 454-461).

Minor

comments:

Line 98-103: The statements sounded like *srf3* mutants are more sensitive to low iron levels, but the data suggest that *srf3-3* is less sensitive to low iron in terms of both primary and secondary root growth. Please consider rephrasing the statements.

The text has been modified accordingly. Thanks.

Are *srf3* mutants contain higher intercellular Fe²⁺?

In our manuscript, we provide evidence that SRF3 senses extracellular signal(s) which are associated with a fast decrease of iron levels. However, we can't establish with certainty if the signal is emitted by a decrease of Fe²⁺ or Fe³⁺ pools that might be intracellular or extracellular. RhoNox-1 is a fluorescent probe which detects the labile Fe²⁺ which is involved in the Fenton reaction to decrease ROS (Fe²⁺ + H₂O₂ → Fe³⁺ + OH· + OH⁻). This probe has been characterized to localized in the cytoplasm, golgi and ER in rat retinal carcinoma^{1,2}. In line with this, we don't think that most of the RhoNox-1 reported signal is apoplasmic as RhoNox-1 does not accumulate in the apoplast in the elongation zone when doing colocalization with a

PM marker (See below). We observed in the merged picture only purple color at the PM instead of white which should raise colocalization between purple and green channels. We used PM marker, because at this confocal resolution we could not differentiate between the PM and apoplast markers.

Regardless of the source of the measured Rho-Nox1 signal, we think that Fe^{2+} is indeed present in plant apoplast since according to the literature H_2O_2 , the Fenton reaction and iron reduction by FRO2 occur in the apoplast/extracellular matrix³⁻⁵. Moreover, apoplastic iron can be detected by ICP-MS⁶.

Left: RhoNox-1. Middle: PM marker. Right: Merge.

Line 139-140: 1. Can *srf3-3* mutant tolerant longer-term iron deficiency during different stages of plant development? 2. As *srf3* mutant was not included in the Figure 5SE, it's unable to determine whether SRF3-OX behaves differently to *srf3* during the late response to low iron.

Compared to the WT, we have demonstrated that SRF3-OX is more sensitive to low iron levels and that *srf3* mutants are less sensitive after long term low iron treatment for 3 days (Figure 1c, Supplementary Figure 1f-i, 5f and 8d).

Due to the clear involvement of SRF3 in the immediate response to low iron levels, we did not systematically conduct experiments at all different developmental stages for other iron related responses. We think in addition to the enormous work it would require, it would further complicate the manuscript (the manuscript is already very dense and contains an enormous amount of data) and detract from the mechanisms that relate to the early responses. However, we obtained some data for this reviewer. In short, *srf3* mutants are more sensitive to low iron than the WT or mutants of the gene adjacent to SRF3 (AT4G03400).

Method:

Alkaline soil was prepared by addition of calcium oxide to a final soil pH of 7.5–8.0 and seeds were stratified for 5 days. Plants were grown in long day conditions (16/8h) in walk in growth chambers (Conviron, Winnipeg, Manitoba, Canada) at 21°C, 50uM light intensity, 60% humidity. During nighttime, temperature was decreased to 15°C. Environmental conditions were established and monitored with commercial software (Valoya, Helsinki, Finland).

Chlorophyll content. 6 days old seedlings were grown on 1/2 MS media were transferred to liquid 1/2 MS in 6 well plates (+Fe (Control), -Fe and -Fe +100uM Ferrozine). 7 days post treatment seedlings were imaged and shoot were cut out to perform chlorophyll assay. Shoots were cut from the roots below the hypocotyl. 5 shoots from the plants grown on a single plate were pooled together to obtain one biological replicate. The shoots were gently dried with tissue paper to remove moisture and weighed. Chlorophyll was measured as described previously¹¹. In brief, leaf tissue from the Arabidopsis seedlings were placed in a vial containing 2 ml dimethylsulfoxide (DMSO) and incubated for 30 min at 65°C. After cooling to RT, total chlorophyll extracts were transferred to a cuvette, and spectrophotometer readings were performed using a spectrophotometer at a wavelength of 645 nm and 663 nm. Chlorophyll content was calculated using Arnon method as described by Hiscox and Israelstam (1979)¹¹.

Line 184-186: The data only support the statement that the extracellular and kinase domains

of SRF3 is required for its function. To demonstrate the fine regulation of SRF3 proteins accumulation at the PM, it is required to demonstrated that misregulation of functional SRF3 at the PM results in the root growth response to low ion conditions.

Thanks for this remark, the text has been edited accordingly.

Figure 3B: The labeling of the figure suggests that PDLP3 and CALS3 are PM markers.

The figure has been edited accordingly. Thanks.

Line 237-238: Please rephrase the subtitle.

The phrase has been modified accordingly. Thanks.

Line 241-245: Do Fen and Lova prevent the PD localization of SRF3? Also, the findings here can only suggest that SRF3 associated to PD through sterol not its functional role in PD.

Because we didn't analyze the Fen or Lova treatments using TEM, we were not able to directly measure this. However, based on the analysis of SDMI its most likely that sterol depletion by Lova and Fen removed SRF3 from the PD. The text has been modified accordingly (Line 258-261). Thanks.

Line 261: add (Figure S4B) following synthase inhibitor.

Callose synthase inhibitor was not used in this experiment. The text has been modified accordingly to avoid confusion. Thanks.

Line 327-329: The statement is unclear. Please consider rephrasing the sentence.

The phrase has been modified accordingly. Thanks.

Line 307: SRF3-OXxcal3-3d > SRF3-OXxcals3-3d

It has been modified accordingly. Thanks.

Line 403-404: The fact that flg22 and low iron treatment doesn't change the SRF3 transcript level is not a supporting evidence for the role of SRF3 in signal transduction.

The sentence has been modified accordingly. Thanks.

Line 430: It's unclear what a signaling function of callose synthases means?

By signaling function, we meant that the canonical function of callose synthases mediating callose deposition is not involved in the SRF3-dependent signaling events which implies that callose synthase signal by an independent mechanism. This point has been clarified throughout the text (line 327-330).

Reviewer #3 (Remarks to the Author):

In this manuscript, the authors identified a receptor kinase gene SRF3 that was associated with the root growth under low iron levels through GWAS. They further provided experimental evidence for the role and mechanism of SRF3 in coordinating root growth, iron homeostasis and immunity pathways. Overall, this manuscript presents an interesting story. However, several issues are also required for major revision. 1. The method of GWAS was used to detected the genes associated with root growth rate under low iron levels, and some SNPs were identified. Because the difference of phenotypic traits among genotypes is the base for GWAS, the authors should firstly illustrate that the population panel is suitable for GWAS analysis.

Similar population panels (200-300 diverse lines from the RegMap panel) were used in previous studies in which GWAS approaches were successfully applied to identify causal genes, e.g. Meijon et al. 2014, Nat. Genetics; Kisko et al., 2018, Elife; Bouain et al. 2018, PLoS Genetics; Ogura et al., 2019, Cell; Li et al. 2019, Nat. Communications)¹²⁻¹⁶. We included this information into the GWAS methods part.

2. More information of GWAS analysis should be provided. The mixed linear model is used for GWAS analysis, according to the description in the section of methods. However, the population structure and kinship should also be provided. In addition, the result of association analysis is suggested to have high probability of false positive, please provide the methods in reducing false positive, for example, the p value threshold.

To clearly provide these informations, we rephrased our methods section for GWA mapping: *"Population structure corrected GWA mapping was conducted using AMM with the 250K SNP dataset (by using the GWAS pipeline implemented on <https://gwas.gmi.oeaw.ac.at/>)¹⁷. For population structure control, this webservice uses an identity by state (IBS) genetic relatedness matrix, first a priori for the full genotype data set and then by removing the contributions of SNPs of accessions that are not contained in the GWAS trait dataset. SNPs with minor allele counts equal or greater to 10 were taken into account. To control for false positives, we used a 5% FDR threshold calculated by the Benjamini-Hochberg-Yekutieli method to correct for multiple testing¹⁸."*

3. Association analysis is based on linkage disequilibrium (LD). The size of the mapping

interval depends on the ratio of LD decay. All the genes in the LD decay interval should be taken into account. To confirm the candidate gene, all the other genes in the interval of LD decay should be excluded. In this manuscript, the T-DNA mutant was used to confirm the causal gene. However, the interval of LD decay and the genes in this interval should be provided. The detailed methods for excluding other genes should also be provided.

We thank the reviewer for this remark. We would like to emphasize that we used the GWAS a screening tool in this manuscript and focused on a novel regulator and mechanism of the root growth response to low iron conditions. We now make more abundantly clear in the text that we can't conclude that the SRF3 alleles are causal for the observed phenotypes in natural accessions and need additional investigation.

4. Results of significance tests should be provided, including the figures 3 and 6A. Please check the raw data and provide clear statistical results for all statistic data.

The test has been modified accordingly. Thanks.

5. The manuscript should be further edited. Many sentences are too long, and it is very hard to get the important information.

We tried to address these issues.

6. It would be nice to provide a diagram for SRF3 to conclude its role and mechanism in the section of discussion.

Thanks for this suggestion. The manuscript already includes an enormous amount of figures and supplemental figures. We therefore felt that we didn't want to add another figure if not strictly necessary.

Specific

issues:

Introduction:

Lines 36-38: Where is the reference? Thanks. It has been modified.

Lines 38-41: This sentence is too long, and hard to understand. Please change. Thanks. It has been modified.

Lines 48-49: The reference should be provided. Thanks. It has been modified.

Line 72: "is" -> "has been" Thanks. It has been modified.

Results:

Lines

92-105:

I noticed that only one SNP with $-\log_{10}(\text{p-value})$ above 6 was shown in the black box of Manhattan plot in both the Figure 1A and S1B, but five SNPs in the magnified regions of

Figure 1A. Please check the GWAS results, and confirm how many SNPs were identified to be associated with root growth rate under low iron levels. In addition, the percentage of phenotypic variations that total and candidate SNPs can explain should be provided, respectively.

Due to the technical limits of the plotting a genome wide Manhattan plot (plotting SNPs over the entire genome in one figure), SNPs are collapsed in the genome wide view if they are of a similar magnitude. It is because of that why we provide the detailed view in which all SNPs in that region are visible. To address this further, we now have provided a table with all the significant SNPs (5% FDR), the % of contribution explained in the model and the adjacent genes (Table S1).

Figure 1A showed four candidate genes in the association regions on chromosome 4. Why did you only investigate two genes *SRF3* and *DLF2*? How did you exclude the other two candidate genes? $-\log(\text{p-value})$ in Figure 1A should be $-\log_{10}(\text{p-value})$.

We thank the reviewer for noting the error in the plot labelling. Figure 1A has now been modified $-\log_{10}(\text{p-value})$.

We chose to investigate only *SRF3* and *DLF2* because the most significant peak was located in between these two genes. We then tested through T-DNA lines the involvement of these two genes in regulating the root growth under low iron levels and observed that only *srf3* mutants showed a phenotype (Figure S1D-I). At this point we focused on the novel mechanism that *SRF3* provided and on experiments to test the *SRF3* involvement. Unfortunately, the experiments testing the involvement of *SRF3* alleles were not fully conclusive and therefore we now stated more clearly in the text, we can't conclude that *SRF3* is the causal gene for the observed natural variation and other genes might be involved.

Line 97: Please show the statistical results for the comparison of the relative expression levels between T-DNA mutant lines and WT in Figure S1D.

It has been added accordingly. Thanks.

Lines 106-111: The statistical results in Figure 1D should be provided. To validate the expression profile from the RNA-seq data, qRT-PCR analysis is also required for examining the transcript levels of these DEGs.

The statistical results are now provided in Figure 1D.

To validate the RNAseq, we have performed qRT-PCR on Flg22 Receptor like Kinase 1 (FRK1, AT2G19190) which is up-regulated in flg22 treatment for 2h and as well MYB51 (AT1G18570) which up regulated under low iron levels and flg22 for 2 hours. (Figure S10C).

Lines 113-116: I cannot follow the relationship between OPT3 and SRF3. RNAseq data also showed that the expression of OPT3 was up-regulated in srf3 mutant?

Opt3-2 and bts-1 are two known mutants which accumulate more iron in roots. See reference¹⁵⁻¹⁷. We used the two mutants as a positive control to show that RhoNox-1 staining is able to detect more iron. We did not notice any significant increase of OPT3 expression in the RNAseq data (Table S2).

Explain.

Lines 153-154: "SRF3 transcript might expressed transiently in the meristematic cells." -> "SRF3 might express transiently in the meristematic cells".

It has been modified accordingly. Thanks.

Lines 160-164: This sentence is too long, please separate it into two sentences.

It has been modified accordingly. Thanks.

Line 257: The reference should be provided for CALS3.

It has been added accordingly. Thanks.

Lines 275-278: Please check the ANOVA results. A decrease of signal in the activated cell and a concomitant increase in the surrounding cells were observed, but your statistical results in Figure 4D showed no differences among all comparisons.

Thanks for pointing this out. This was a mistake while generating the figure. The test has been performed properly and the results are now represented in Figure 4d. See result test below.

Geno-cond / Fisher (LSD) / Analysis of the differences between the categories with a confidence interval of 95% (Value):					
Contrast	Difference	Standardize d difference	Critical value	Pr > Diff	Significant
srf3-3_Around_+Fe vs srf3-3_Activated_+Fe	0.239	3.844	1.975	<0.0001	Yes
srf3-3_Around_+Fe vs WT_Activated_+Fe	0.232	4.063	1.975	<0.0001	Yes
srf3-3_Around_+Fe vs srf3-3_Activated_-Fe	0.193	2.909	1.975	0.000	Yes
srf3-3_Around_+Fe vs WT_Activated_-Fe	0.159	2.570	1.975	0.002	Yes
srf3-3_Around_+Fe vs WT_Around_-Fe	0.159	2.570	1.975	0.002	Yes
srf3-3_Around_+Fe vs WT_Around_+Fe	0.106	1.745	1.975	0.038	Yes
srf3-3_Around_+Fe vs srf3-3_Around_-Fe	0.084	1.251	1.975	0.110	No
srf3-3_Around_-Fe vs srf3-3_Activated_+Fe	0.155	3.056	1.975	0.003	Yes
srf3-3_Around_-Fe vs WT_Activated_+Fe	0.148	3.337	1.975	0.002	Yes
srf3-3_Around_-Fe vs srf3-3_Activated_-Fe	0.109	1.955	1.975	0.028	Yes
srf3-3_Around_-Fe vs WT_Activated_-Fe	0.075	1.490	1.975	0.106	No
srf3-3_Around_-Fe vs WT_Around_-Fe	0.075	1.490	1.975	0.106	No
srf3-3_Around_-Fe vs WT_Around_+Fe	0.022	0.453	1.975	0.640	No
WT_Around_+Fe vs srf3-3_Activated_+Fe	0.133	3.153	1.975	0.010	Yes
WT_Around_+Fe vs WT_Activated_+Fe	0.126	3.681	1.975	0.007	Yes
WT_Around_+Fe vs srf3-3_Activated_-Fe	0.087	1.807	1.975	0.069	No
WT_Around_+Fe vs WT_Activated_-Fe	0.053	1.268	1.975	0.235	No
WT_Around_+Fe vs WT_Around_-Fe	0.053	1.268	1.975	0.235	No
WT_Around_-Fe vs srf3-3_Activated_+Fe	0.080	1.840	1.975	0.107	No
WT_Around_-Fe vs WT_Activated_+Fe	0.073	2.036	1.975	0.101	No
WT_Around_-Fe vs srf3-3_Activated_-Fe	0.034	0.692	1.975	0.459	No
WT_Around_-Fe vs WT_Activated_-Fe	0.000	0.000	1.975	1.000	No
WT_Activated_-Fe vs srf3-3_Activated_+Fe	0.080	1.840	1.975	0.107	No
WT_Activated_-Fe vs WT_Activated_+Fe	0.073	2.036	1.975	0.101	No
WT_Activated_-Fe vs srf3-3_Activated_-Fe	0.034	0.692	1.975	0.459	No
srf3-3_Activated_-Fe vs srf3-3_Activated_+Fe	0.046	0.923	1.975	0.380	No
srf3-3_Activated_-Fe vs WT_Activated_+Fe	0.039	0.900	1.975	0.413	No
WT_Activated_+Fe vs srf3-3_Activated_+Fe	0.007	0.193	1.975	0.890	No
LSD-value:			0.084		

Category	LS means	Standard error	Lower bound (95%)	Upper bound (95%)	Groups
srf3-3_Around_+Fe	1.126	0.039	1.049	1.203	A
srf3-3_Around_-Fe	1.042	0.035	0.974	1.111	A
WT_Around_+Fe	1.020	0.032	0.956	1.084	B
WT_Activated_-Fe	0.967	0.030	0.908	1.027	B
WT_Around_-Fe	0.967	0.030	0.908	1.027	B
srf3-3_Activated_-Fe	0.933	0.035	0.865	1.002	C
WT_Activated_+Fe	0.894	0.032	0.831	0.958	D
srf3-3_Activated_+Fe	0.887	0.039	0.810	0.964	D

Lines 292-297: The ANOVA results for Figure 5B should be provided.

It has been added accordingly. Thanks.

Line 303: Please provide the reference for "PdBG1 known to negatively regulate callose deposition."

It has been added accordingly. Thanks.

Line 307. Please clearly point out the SRF3-OXxcal3-3d line in the picture of Figure 5D.

It has been modified accordingly for more clarity. Thanks.

Lines 327-329: What is the role of SRF3 in roots? Please clearly state the results from GO

analysis of RNAseq data and GUS staining assays, respectively, and point out the role of SRF3 in roots here. In addition, the ANOVA analysis for GUS quantification should be performed.

It has been modified accordingly for more clarity. Thanks. (Line 355-359; Figure 6a). A two ways student test has been done on GUS quantification ($p < 0.05$)

Lines 332-333: Figure S8B showed that the difference of late root growth response to flg22 was not significant between WT and srf3-3, so how did you conclude that srf3 roots were only impaired in their response to flg22? Please re-analyze the raw data or revise this result.

Thanks for the suggestion. It was a mistake and we have corrected it (see ANOVA results below).

Line 337: "higher iron root content" -> "higher iron content in the root"
It has been modified accordingly. Thanks.

Lines 352-353: The ANOVA results for Figure 6E should be provided.

It has been added accordingly. Thanks.

Discussion:

The authors spent much time on concluding and repeating the results. It would be nice to point out and discuss several important novel insights of this manuscript. English writing has to be improved. More importantly, can you provide a diagram for SRF3 to explain its role and mechanism in coordinating root growth, iron homeostasis and bacterial immune responses?

We thank the reviewer for their suggestion and have tried to be more concise and improve the writing. We were not convinced that adding another figure with a model would improve the manuscript.

Methods

Lines 717-730: Please provide more detailed methods for GWAS analysis.

We now have expanded our GWAS method section to provide more detail and clarity.

Line 794: I could not find the reference for Platre et al, 2019.

It has been added accordingly. Thanks.

Lines 832, 888, 895 and 976: Please provide the reference for Gruber et al., 2013.

It has been added accordingly. Thanks.

Lines 904 and 907: The reference for Roschztardt et al., 2009 and Stacey et al., 2008, should be provided, respectively.

It has been added accordingly. Thanks.

Lines 1133 and 1142: Please provide the reference for Vert et al., 2002 and Nakagawa et al, 2007, respectively.

It has been added accordingly. Thanks.

1. Mukaide, T. *et al.* Histological detection of catalytic ferrous iron with the selective turn-on fluorescent probe RhoNox-1 in a Fenton reaction-based rat renal carcinogenesis model. *Free Radical Research* **48**, 990–995 (2014).

2. Carter, K. P., Young, A. M. & Palmer, A. E. Fluorescent Sensors for Measuring Metal Ions in Living Systems. *Chem. Rev.* **114**, 4564–4601 (2014).
3. Connolly, E. L., Campbell, N. H., Grotz, N., Prichard, C. L. & Guerinot, M. L. Overexpression of the FRO2 ferric chelate reductase confers tolerance to growth on low iron and uncovers posttranscriptional control. *Plant Physiol* **133**, 1102–1110 (2003).
4. Martinière, A. *et al.* Osmotic Stress Activates Two Reactive Oxygen Species Pathways with Distinct Effects on Protein Nanodomains and Diffusion. **179**, 13 (2020).
5. Smirnoff, N. & Arnaud, D. Hydrogen peroxide metabolism and functions in plants. *New Phytologist* **221**, 1197–1214 (2019).
6. Nobori, T. *et al.* Transcriptome landscape of a bacterial pathogen under plant immunity. *PNAS* **115**, E3055–E3064 (2018).
7. Cui, Y. *et al.* Four IVa bHLH Transcription Factors Are Novel Interactors of FIT and Mediate JA Inhibition of Iron Uptake in Arabidopsis. *Molecular Plant* **11**, 1166–1183 (2018).
8. Ye, L. *et al.* MPK3/MPK6 are involved in iron deficiency-induced ethylene production in Arabidopsis. *Frontiers in Plant Science* **6**, 953 (2015).
9. Lingam, S. *et al.* Interaction between the bHLH transcription factor FIT and ETHYLENE INSENSITIVE3/ETHYLENE INSENSITIVE3-LIKE1 reveals molecular linkage between the regulation of iron acquisition and ethylene signaling in Arabidopsis. *Plant Cell* **23**, 1815–1829 (2011).

10. Vatén, A. *et al.* Callose biosynthesis regulates symplastic trafficking during root development. *Dev Cell* **21**, 1144–1155 (2011).
11. Hiscox, J. D. & Israelstam, G. F. A method for the extraction of chlorophyll from leaf tissue without maceration. *Can. J. Bot.* **57**, 1332–1334 (1979).
12. Meijón, M., Satbhai, S. B., Tsuchimatsu, T. & Busch, W. Genome-wide association study using cellular traits identifies a new regulator of root development in Arabidopsis. *Nat Genet* **46**, 77–81 (2014).
13. Kisko, M. *et al.* LPCAT1 controls phosphate homeostasis in a zinc-dependent manner. *eLife* **7**, e32077 (2018).
14. Bouain, N. *et al.* Systems genomics approaches provide new insights into Arabidopsis thaliana root growth regulation under combinatorial mineral nutrient limitation. *PLOS Genetics* **15**, e1008392 (2019).
15. Ogura, T. *et al.* Root System Depth in Arabidopsis Is Shaped by EXOCYST70A3 via the Dynamic Modulation of Auxin Transport. *Cell* **178**, 400–412.e16 (2019).
16. Li, B. *et al.* GSNOR provides plant tolerance to iron toxicity via preventing iron-dependent nitrosative and oxidative cytotoxicity. *Nature Communications* **10**, 13 (2019).
17. Seren, Ü. *et al.* GWAPP: A Web Application for Genome-Wide Association Mapping in Arabidopsis. *The Plant Cell* **24**, 4793–4805 (2012).
18. Benjamini, Y. & Yekutieli, D. The control of the false discovery rate in multiple testing under dependency. *Ann. Statist.* **29**, 1165–1188 (2001).

REVIEWER COMMENTS

Reviewer #1 (Remarks to the Author):

I enjoyed reading the revised manuscript, which is now substantially improved. The authors have responded to all my comments and successfully addressed my concerns by including new experimental data or additional explanation in the text. This is a very impressive study with several important and novel messages. I have no further comments to add.

Reviewer #2 (Remarks to the Author):

The authors made a major effort with the submission. Many more data were added to a large body of data already presented in the first submission.

SRF3 seems to be an important receptor kinase, regulating several aspects of plant cellular processes. The current version also puts callose synthases at the center for SRF3 in coordinating root growth, iron homeostasis and immunity. It's highly likely that callose accumulation is one of the cellular processes regulated by SRF3.

From the provided evidence, it's clear that SRF3 does not regulate the plasmodesmal function. The data presented in Figure 4 mostly suggest that SRF3 does not affect callose accumulation at PD. It is possible that SRF3 regulates callose accumulation in the apoplast as likely supported by Figure 4a. Further research is needed to establish the role of SRF3 in modulating callose biosynthesis.

The hypersensitive and hyposensitive to low iron in *cals3-6* and *CalS3-OX*, respectively, provide more supporting evidence on the role of callose biosynthesis on root growth under low iron levels.

The hyposensitive phenotype to low iron of *SRF3-OX cals3-6* is intriguing, but the finding does not suggest the genetic interaction between the two genes.

Out of 12 *Arabidopsis* CalSs, it's unclear how the authors narrowed it down to *CalS3*. The characterization of *cals3-3d* is logical as it's a gain of function mutant. Does RNAseq show any CalSs or β -1,3-Glucanases differentially expressed in *srf3* mutant?

If callose synthase is responsible for iron homeostasis and plant immunity, does *CalS3-OX* exhibit lower pIRT1 activity under a low iron condition similar to *srf3* mutants? Also, is *CalS3-OX* root growth hyposensitive to *flg22* treatment similar to *srf3* mutant?

The authors coined a new term "Callose synthase signaling." Despite the explanation, it still sound a bit abstract. It seems to suggest callose biosynthesis independent function of CalSs rather than a signaling pathway.

Line 297-299: There is no evidence to support the statement that "an early decrease of iron levels swift leads to callose synthase-dependent modulation of callose deposition." In order to make the statement, iron insensitive phenotype of *cals* knock out mutants is necessary. The result cannot rule out the possibility that callose overaccumulate due to the suppression of β -1,3-Glucanases or stabilization of callose.

Line 319: The experiment does not determine the interaction between SRF3 and callose synthases.

Line 327-330: The finding that SRF3 overexpression does not suppress the *cals3-3d* root growth phenotype is not sufficient to suggest a direct role of SRF3 in *CalS3*-mediated signaling.

Line 330-332: Using PDL5-OX to uncouple the increase of callose from the signaling activity of cal3-3d seems unusual. Callose overaccumulation phenotype in PDL5-OX is likely dependent on CalSs. Not sure what the authors meant by "which represents identical root growth and callose deposition phenotype." Comparing between cal3-3d and PDL5-OX or SRF3-OX and PDL5-OX?

Line 336: "no callose deposition" sounds like cal3-6 mutant lacks callose deposition.

To determine the genetic interaction between SRF3 and CalS3, it would be important to characterize srf3 cal3 double mutant. If the double mutant becomes hypersensitive to iron deficiency, one could conclude the genetic interaction between the two genes.

Line 341-343: If CalS3 acts downstream of SRF3, would overexpression of SRF3 suppresses the function of CalS3? The transgenic plants overexpressing both genes seem to suggest that SRF3 has no effect on CalS3.

Line 395-397: The sentence is a bit confusing. From the data, ~90% of DEGs under -Fe overlapped with less than 10% of DEG under flg22 treatment.

Reviewer #3 (Remarks to the Author):

The authors have addressed all my concerns.

We thank the reviewers for taking the time to evaluate our revised manuscript. We have addressed all points that were raised for the revised manuscript by reviewer 2.

REVIEWER

COMMENTS

Reviewer #1 (Remarks to the Author):

I enjoyed reading the revised manuscript, which is now substantially improved. The authors have responded to all my comments and successfully addressed my concerns by including new experimental data or additional explanation in the text. This is a very impressive study with several important and novel messages. I have no further comments to add.

We thank the reviewer for this assessment.

Reviewer #2 (Remarks to the Author):

The authors made a major effort with the submission. Many more data were added to a large body of data already presented in the first submission.

SRF3 seems to be an important receptor kinase, regulating several aspects of plant cellular processes. The current version also puts callose synthases at the center for SRF3 in coordinating root growth, iron homeostasis and immunity. It's highly likely that callose accumulation is one of the cellular processes regulated by SRF3.

We thank the reviewer for this assessment.

From the provided evidence, it's clear that SRF3 does not regulate the plasmodesmal function. The data presented in Figure 4 mostly suggest that SRF3 does not affect callose accumulation at PD. It is possible that SRF3 regulates callose accumulation in the apoplast as likely supported by Figure 4a. Further research is needed to establish the role of SRF3 in modulating callose biosynthesis.

The hypersensitive and hyposensitive to low iron in *cals3-6* and *CalS3-OX*, respectively, provide more supporting evidence on the role of callose biosynthesis on root growth under low iron levels.

The hyposensitive phenotype to low iron of *SRF3-OX cals3-6* is intriguing, but the finding does not suggest the genetic interaction between the two genes.

We have modified the text accordingly (lines 342-348):" *cals3-6 was more sensitive to low iron compared to the WT, thus mimicking the phenotype of the SRF3-OX line and therefore further indicating that CalS3 is required to modulate root growth under low iron (Fig. 5e). Surprisingly, in the double mutant cals3-6/SRF3-OX, root growth was less sensitive to low iron conditions compared to WT, supporting that these two genes regulate root growth under low iron conditions (Fig. 5e). Because no additive hypersensitivity was observed (Fig. 5e), these data suggested that they might be involved in the same pathway.*"

Out of 12 Arabidopsis CalSs, it's unclear how the authors narrowed it down to *CalS3*. The characterization of *cals3-3d* is logical as it's a gain of function mutant. Does RNAseq show any CalSs or β -1,3-Glucanases differentially expressed in *srf3* mutant?

We didn't find any CalSs or β -1,3-Glucanases in the RNAseq that were differentially expressed in *srf3* mutants. As aforementioned, we initially utilized *CalS3* because of the gain of function mutant *cals3-3d*. This yielded in an unexpected but informative result: when we overexpressed *SRF3* in *cals3-3d*, we didn't observe a suppression of the *cals3-3d* root growth defects but a further increase. Moreover, we felt encouraged to focus on *CalS3* because of its expression in the root tip, which is similar to *SRF3* (meristem, transition/elongation zone) (Vaten et al., 2011, Dong et al., 2008 Planta, single cell RNAseq). Finally, genetic material to study *CalS3* function has been developed, *cals3-5*, *cals3-6*, *CALS3-OX* and *cals3-3d*, enabling us to study its function in more detail.

If callose synthase is responsible for iron homeostasis and plant immunity, does *CalS3-OX* exhibit lower pIRT1 activity under a low iron condition similar to *srf3*

mutants? Also, is CalS3-OX root growth hyposensitive to flg22 treatment similar to srf3 mutant?

In the current manuscript we have provided data showing that upon low iron or flg22 while treating with DDG to inhibit callose activity (which the latter treatment does not influence the iron content in root, Fig. S8a) the expression of IRT1 is not as strong as compared to mock condition in low iron or flg22 conditions respectively (Fig. 5b and 6e). Thus, this provides some evidence of the role of callose synthases activity in these processes. Moreover, we did not indicate that this process is specific of CalS3. All together, we do not think that providing more data on this aspect is necessary given the scope of our study that is mainly focused on the receptor kinase SFR3.

The authors coined a new term "Callose synthase signaling." Despite the explanation, it still sounds a bit abstract. It seems to suggest callose biosynthesis independent function of CalSs rather than a signaling pathway.

Thanks for this remark and have revised the manuscript accordingly.

Line 297-299: There is no evidence to support the statement that "an early decrease of iron levels swift leads to callose synthase-dependent modulation of callose deposition." In order to make the statement, iron insensitive phenotype of cals knock out mutants is necessary. The result cannot rule out the possibility that callose overaccumulate due to the suppression of β -1,3-Glucanases or stabilization of callose.

We do think that our results support the involvement of CalS proteins in the process studied: First, PDBG1 seems not to colocalize at the PM with SRF3 while CalS3 does, using TIRF microscopy (Fig. 5c). Second, while using DDG to inhibit callose activity no increase of callose deposition was detected under low iron according to aniline blue staining (Fig S7b). Finally, application of DDG partially complement the early and late root growth defect of srf3-3 under low iron (Fig. 4g and S8d). We therefore revised the text (Line 296-299) : *"Altogether, our results suggest that an early decrease of iron levels swiftly leads to SRF3-*

dependent modulation of callose deposition which might involve callose synthases, however, this does not generally impede cell-to-cell movement.”.

Line 319: The experiment does not determine the interaction between SRF3 and callose synthases.

Thank you. The text has been modified accordingly (line 319-320): *“We then investigated whether SRF3 and callose synthases can co-exist in the same region of the plasma membrane in roots”*

Line 327-330: The finding that SRF3 overexpression does not suppress the cal3-3d root growth phenotype is not sufficient to suggest a direct role of SRF3 in CalS3-mediated signaling.

Thank you. The text has been edited (line 329-334) *“Since ectopic expression of SRF3 did not suppress the cal3-3d root growth phenotype, these data support a model in which SRF3 acts together with Cals3 to regulate root growth. However, this phenomenon seems to be independent of callose synthases-mediated callose biosynthesis because SRF3 overexpression, acting as a negative regulator of callose deposition, was not able to recover callose overaccumulation-related cal3-3d root growth phenotype.”*

Line 330-332: Using PDLP5-OX to uncouple the increase of callose from the signaling activity of cal3-3d seems unusual. Callose overaccumulation phenotype in PDLP5-OX is likely dependent on CalSs. Not sure what the authors meant by “which represents identical root growth and callose deposition phenotype.” Comparing between cal3-3d and PDLP5-OX or SRF3-OX and PDLP5-OX?

We apologize for the confusion. This section has been edited for more clarity as follow: (line 334-340): *“To test the independence of callose biosynthesis, we crossed SRF3-OX with 35s::GFP-PDLP5 (PDLP5-OX) line. PDLP5-OX displays root growth and callose deposition phenotypes similar to cal3-3d as PDLP5 most likely acts as a positive regulator of CalS1 and CalS8-mediated callose biosynthesis³⁸⁻⁴⁰. The root growth phenotype of this cross was indistinguishable*

from the PDLP5-OX line, supporting the model that SRF3 might control root growth with CalS3 independently of its callose biosynthesis activity (Supplementary Fig. 8b)."

Line 336: "no callose deposition" sounds like cals3-6 mutant lacks callose deposition.

The text has been modified accordingly (line 341-342): "This line displays neither root growth nor callose accumulation defects in standard growth conditions³⁰."

To determine the genetic interaction between SRF3 and CalS3, it would be important to characterize srf3 cals3 double mutant. If the double mutant becomes hypersensitive to iron deficiency, one could conclude the genetic interaction between the two genes.

We agree that this experiment would be required to making a strong and unambiguous conclusion of SRF3/CalS3. This will be very interesting a future work. To make sure not to overstate, we phrased our conclusion as follow (line 360-363): "In sum, our data suggest that SRF3 acts along with callose synthases early-on upon low iron levels to regulate iron homeostasis and root growth, however, we were not able to directly decipher the sequentially of the pathway hinting towards a complex functional interaction of SRF3 and CalSs."

Line 341-343: If CalS3 acts downstream of SRF3, would overexpression of SRF3 suppresses the function of CalS3? The transgenic plants overexpressing both genes seem to suggest that SRF3 has no effect on CalS3.

We used the CALS3-OX and SRF3-OX which both have ectopic overexpression outside of their native expression pattern so it's hard to conclude about which one is upstream. Thus, we therefore broadly concluded that SRF3 and CalS3 act on the same pathway and SRF3 might act downstream of CalS3. The text has been edited as follow (line 348-353): "To further test this, we made use of the CALS3-OX line, which is less sensitive to low iron and therefore presents the opposite response compared to SRF3-OX in reference to WT (Fig. 5f). Using these lines, we generated the double mutant CalS3-OX/SRF3-OX which behaved like

CalS3-OX compared to WT, indicating that CalS3 and SRF3 are both involved in the same pathway and that SRF3 might act downstream of CalS3 (Fig. 5f)."; However, the next experiments, using SRF3 cellular behavior and root growth as a read out upon DDG suggested the opposite (Fig. 5g and sS8c-d). Thus, we concluded that in line 362-363 "we were not able to directly decipher the sequentially of the pathway hinting towards a complex functional interaction of SRF3 and CalSs."

Line 395-397: The sentence is a bit confusing. From the data, ~90% of DEGs under -Fe overlapped with less than 10% of DEG under flg22 treatment.

The phrase has been changed as follows (line 407-409):" Strikingly, in WT, 90% of the differentially expressed genes (DEGs) in low iron overlapped with DEGs upon flg22 treatment and were regulated in the same manner (e.g. up-regulated DEGs in low iron were also up-regulated upon flg22)."

Reviewer #3 (Remarks to the Author):

The authors have addressed all my concerns.

We thank the reviewer for this assessment.

REVIEWERS' COMMENTS

Reviewer #2 (Remarks to the Author):

The revised version looks great. No further comments.